



**Influence of multi-decadal land use, irrigation practices and climate on riparian corridors**
**across the Upper Missouri River Headwaters Basin, Montana**
Melanie K. Vanderhoof[1], Jay R. Christensen[2], Laurie C. Alexander[3]
[1]US Geological Survey, Geosciences and Environmental Change Science Center, P.O. Box
25046, DFC, MS980, Denver, CO 80225, USA
[2]US Environmental Protection Agency, Office of Research and Development, National Exposure
Research Laboratory, 26 W. Martin Luther King Dr., MS-642, Cincinnati, OH 45268, USA
[3]US Environmental Protection Agency, Office of Research and Development, National Center
for Environmental Assessment, 1200 Pennsylvania Ave NW (8623-P), Washington, DC
20460, USA
Corresponding Author: Melanie K. Vanderhoof (mvanderhoof@usgs.gov, 303.236.1411)
**Abstract**
The Upper Missouri River Headwaters Basin (36,400 km$^2$) depends on its river corridors to
support irrigated agriculture and world-class trout fisheries. We evaluated trends (1984-2016) in
riparian wetness, an indicator of riparian condition, in peak irrigation months (June, July,
August) for 158 km$^2$ of riparian area across the basin using the Landsat Normalized Difference
Wetness Index (NDWI). We found that 8 of the 19 riparian reaches across the basin showed a
significant drying trend over this period, including all three basin outlet reaches along the
Jefferson, Madison and Gallatin Rivers. The influence of upstream climate was quantified using
per reach random forest regressions. Although much of the interannual variability was explained
by climate, especially by drought indices and annual precipitation, the significant drying trends
persisted in the NDWI-climate model residuals, indicating that trends were not entirely
attributable to climate. Over the same period we documented a 506% increase in center-pivot
irrigation and an associated 39% decrease in non-center pivot irrigation basin-wide. Riparian
reaches with a drying trend had a greater shift towards center-pivot irrigation relative to riparian
reaches without such a trend ($p<0.1$). The drying trend, however, did not extend to river
discharge. Over the same period, stream gages (n=7) showed a positive correlation with riparian
wetness ($p<0.05$), but no trend in summer river discharge, suggesting that riparian areas may be
more sensitive to changes in irrigation return flows, relative to river discharge. Identifying trends
in riparian vegetation is a critical precursor to enhancing the resiliency of river systems and
associated riparian corridors.



**Keywords**

Center-pivot, discharge, headwaters, Landsat, precipitation, wetness

**1. Introduction**

Riparian ecosystems provide critical biological, chemical and hydrological functions (Fritz et al., 2018). Defined as semi-terrestrial areas influenced by freshwaters at the interface of rivers and adjacent upland areas (Naiman et al., 2005), riparian ecosystems store water, nutrients, and sediments, reducing downstream flood impacts and non-point source pollution (Lowrance et al., 1984; Vivoni et al., 2006). They also provide corridors for biotic movement and migration, particularly through arid, urban and agricultural landscapes (Boutin and Belanger, 2003; Lees and Peres, 2008), and maintain fish habitat by lowering stream temperatures and contributing in-stream woody debris (Poole and Berman, 2001; Isaak et al., 2012). Long-term trends in the degradation of riparian areas are common globally (Stromberg, 2001; Richardson et al., 2007). The hydrological alteration of rivers, including dam construction, ditching, flow regulation, and pumping of surface and ground water for human use, can alter flow timing and magnitude leading to riparian degradation including changes to riparian functioning, loss of riparian extent, and shift in species composition (Poff et al., 1997; Nilsson and Berggren, 2000; Sweeney et al., 2004). Periodic drought and continued water withdrawals degrade cold-water spawning and rearing habitat for salmonid species (Clancy, 1988; Isaak et al., 2012). Balancing anthropogenic water needs while maintaining or enhancing riparian ecosystem integrity requires an improved understanding of the relationship between water extraction, river discharge, and riparian vegetation (Jones et al., 2010; Cunningham et al., 2011).

Irrigated agriculture is a primary consumptive use of water in the United States and globally. Across the United States, 26% of surface water withdrawals and 68% of groundwater withdrawals are attributable to agricultural irrigation (Dieter et al., 2018). Globally, irrigation accounts for 70% of water withdrawals (Wisser et al., 2008). Expansion of agricultural irrigation over the past centuries and shifts in irrigation methods over the past decades have led to major gains in agricultural productivity, food security, profitability, and crop diversification (Falkenmark and Lannerstad, 2005). As a primary use of water withdrawals and water consumption, however, irrigated agriculture can be expected to play a key role in local water cycles. When gravity-fed (i.e., flood) irrigation is applied, water that is not evaporated or

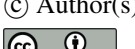



transpired by plants, replenishes soil water storage, recharges aquifers, and contributes return
flows to streams and wetlands (Peterson and Ding, 2005; Perry, 2017; Grafton et al., 2018).
Additional groundwater recharge also comes from unlined ditch systems used to convey water to
agricultural fields. Return flow from excess irrigation has been argued to have artificially
elevated autumn and winter streamflow for decades (Kendy and Bredehoeft, 2006). As farmers
switch to more modern irrigation techniques, such as center pivot irrigation, they can achieve
greater crop yields and gross revenue with less water, improving their "crop per drop" ratio (or
water use efficiency; Peterson and Ding, 2005). This change, however, is also be expected to
have hydrological consequences, namely increased evapotranspiration, and a reduction in surface
runoff and subsurface recharge (Ward and Pulido-Velazquez, 2008; Grafton et al., 2018) which
can impact local aquifers (Peterson and Ding, 2005; Pfeiffer and Lin, 2014), base flow (Kendy
and Bredehoeft, 2006; Gosnell et al., 2007), as well as riparian ecosystems (Carrillo-Guerrero,

2013).

Although water withdrawals for irrigation may impact local water cycling, patterns in

river discharge and riparian vegetation are largely driven by the watershed's climate patterns.
Riparian vegetation tends to be adapted to highly variable fluvial disturbance regimes, a product
of seasonal and interannual variability in river discharge, with riparian wetness peaking during
episodic storm and flood events and lessening during drought events (Hughes, 2005; Goudie,
2006; Capon, 2013). River discharge and groundwater hydrology, in turn, tends to be highly
responsive to variability in precipitation and evaporative demand (Goudie, 2006; Dragoni and
Sukhiga, 2008; Hausner et al., 2018). Further, in snow-melt dominated systems, changes in snow
pack storage and rain to snow event ratios can influence the timing of river discharge and
regional groundwater recharge, impacting water availability in associated riparian areas (Rood et
al., 2008).

While satellite imagery offers a cost-effective means to monitor landscapes, the narrow,

linear nature of riparian corridors presents a challenge for ecosystem characterization with
remote sensing tools (Klemas, 2014; Vanderhoof and Lane, 2019). Along large rivers, Landsat
satellites provide a multi-decadal source of imagery to monitor changes in riparian vegetation
(Jones et al., 2010; Henshaw et al., 2013). Remote sensing can also complement field data to
enhance our understanding of the relationship between riparian vegetation and agents of change,
such as climate (Huntington et al., 2016). The Normalized Difference Vegetation Index (NDVI)



(Tucker, 1979) is the most commonly used spectral index to evaluate changes in riparian
vegetation over time (Fu and Burgher, 2015; Hamdan and Myint, 2015; Nguyen et al., 2015;
Hausner et al., 2018). Trends in riparian greenness have been related successfully to climate
variables and river discharge (Shafroth et al., 2002; Fu and Burgher, 2015; Nguyen et al., 2015),
in part because riparian and wetland herbaceous species can respond rapidly to changes in soil
moisture. Thus, riparian greenness tends to reflect river corridor hydrologic processes
(Stromberg et al., 2001, 2006; Jones et al., 2008). Other indices can also potentially inform
riparian wetness. For instance, the normalized difference wetness index (NDWI) was designed to
be sensitive to changes in leaf and soil water content as well as to identify waters associated with
wetlands or floodplains (Gao, 1996; McFeeters, 1996). This index has been used successfully,
for example, to monitor changes in the extent of waterlogged areas (e.g., Chatterjee et al.,
2005; Chowdary et al., 2008).
Despite the potential for satellite imagery to characterize plant-water interactions along
riparian corridors, few studies have evaluated the impact of changing irrigation methods on
riparian vegetation (Klemas, 2014; Perry, 2017), or have attempted to distinguish the relative
influence of climate and agricultural irrigation on riparian vegetation. The Upper Missouri River
Headwaters (UMH) Basin in southwestern Montana provides an excellent case study for
exploring the interactions between climate, irrigation and riparian vegetation. The basin contains
the Jefferson, Madison, and Gallatin Rivers, all of which support world-class cold-water trout
fisheries that provide substantial economic value to the region (Duffield et al., 1992; Kerkvliet et
al., 2002; Gosnell et al., 2007). In addition, the agricultural valleys of the basin are very
productive yet rely on a complex irrigation system to water crops grown in and near riparian
areas. Irrigation accounts for 97% of Montana's consumptive water use (Clifford, 1995; Dieter et
al., 2018). Along with the high demand for irrigation water (Goklany, 2002; Schaible and
Aillery, 2012), there are also increasing public water supply needs in the basin (Hansen et al.,
2002; Gude et al., 2006). In addition, the timing of peak river flows is predicted to change,
attributable to warmer temperatures at higher elevations and more precipitation in winter and
early spring occurring as rainfall rather than snow (Pederson et al., 2011, 2013; USBR, 2012).
All of these factors are contributing to an increasingly uncertain supply of water across the basin,
particularly in the late summer. This uncertainty, in turn, has elevated interest in improving the
resiliency of local streams and rivers so that the basin can continue to support the agricultural,





recreational, municipal and ecological needs of the watershed (Ziemer et al., 2006; Jones et al.,
2012; Gärtner et al., 2013). In this study we used a time series of Landsat imagery (1984-2016)
together with climate datasets, agricultural datasets, and U.S. Geological Survey (USGS) stream
gage datasets to explore trends over time in riparian vegetation for the major river valleys across
the UMH Basin. We sought to link the temporal trends not explained by climate to changes in
land use type and intensity. Our research questions included:

1. How does remotely sensed riparian wetness across the UMH Basin reflect interannual

variability in climate and river discharge?

2. How and to what degree are trends in riparian wetness from 1984-2016 attributable to

changes in climate versus shifts in land use such as irrigation practice?


**2. Methods**
**2.1 Study Area**

The study area was the UMH Basin (36,400 km$^2$). Near the basin outlet, the Jefferson,

Madison ,and Gallatin Rivers merge to form the Missouri River at Three Forks, Montana. A total
of nine rivers were included in the analysis with riparian vegetation divided into 19 riparian
reaches (Fig. 1). Hydrologic regimes of the rivers across the basin are snow-melt dominated
(Markstrom et al., 2016; Cross et al., 2017) with multiple mountain ranges contributing surface
runoff and ground water recharge to valley aquifers (Hackett et al. 1960; Slagle 1995). Annual
precipitation across the basin averages 565 mm yr$^{-1}$, most of which falls in the mountains, where
it is received primarily as snow (Fig. 2). The annual maximum and minimum temperatures
average 10 °C and −3 °C respectively (1981-2010 period of record) (PRISM Climate Group,
2018). Elevations across the basin range from 1231 m to 3433 m (Gesch, 2002). While the
mountain ranges are dominated by evergreen forest (35%), at lower elevations, the forest gives
way to herbaceous vegetation (35%) and shrub/scrub (20%) cover types that dominate the large
river valleys (Homer et al., 2015, Fig. 2). Agriculture occurs primarily in the lower elevations
adjacent to many of the major rivers. As of 2017, alfalfa was the most common crop (41%),
followed by other non-alfalfa hay crops (25%), barley (11%) and spring wheat (11%) (USDA,
2018). The riparian ecosystems along the major rivers are dominated by tree species including
cottonwood (*Populus* spp.), willow (*Salix* spp.), and alder (*Alnus* spp.); shrubs including
chokecherry (*Prunus virginiana*), snowberry (*Symphoricarpos* spp.), and wild rose (*Rosa*



*woodsia*); and wet meadows dominated by cattails (*Typha* spp.), sedges (*Carex* spp.), and rushes
(*Juncus* spp.). Warming temperatures in March and April initiate snowmelt and a corresponding
increase in river discharge. Spring precipitation and snowmelt produce peak river discharge in
May and June (Cross et al., 2017) followed by a sharp decline in July and August due to a
dwindling supply of melt water from snow pack and consumptive use from withdrawals. Late
autumn through early spring are generally characterized by lower flow conditions, presumably
dominated by baseflow contributions from groundwater discharge (Cross et al., 2017). Major
waterbodies across the basin are predominately reservoirs located upstream from dams (Fig. 1b)
that support irrigation, hydropower, and recreation.



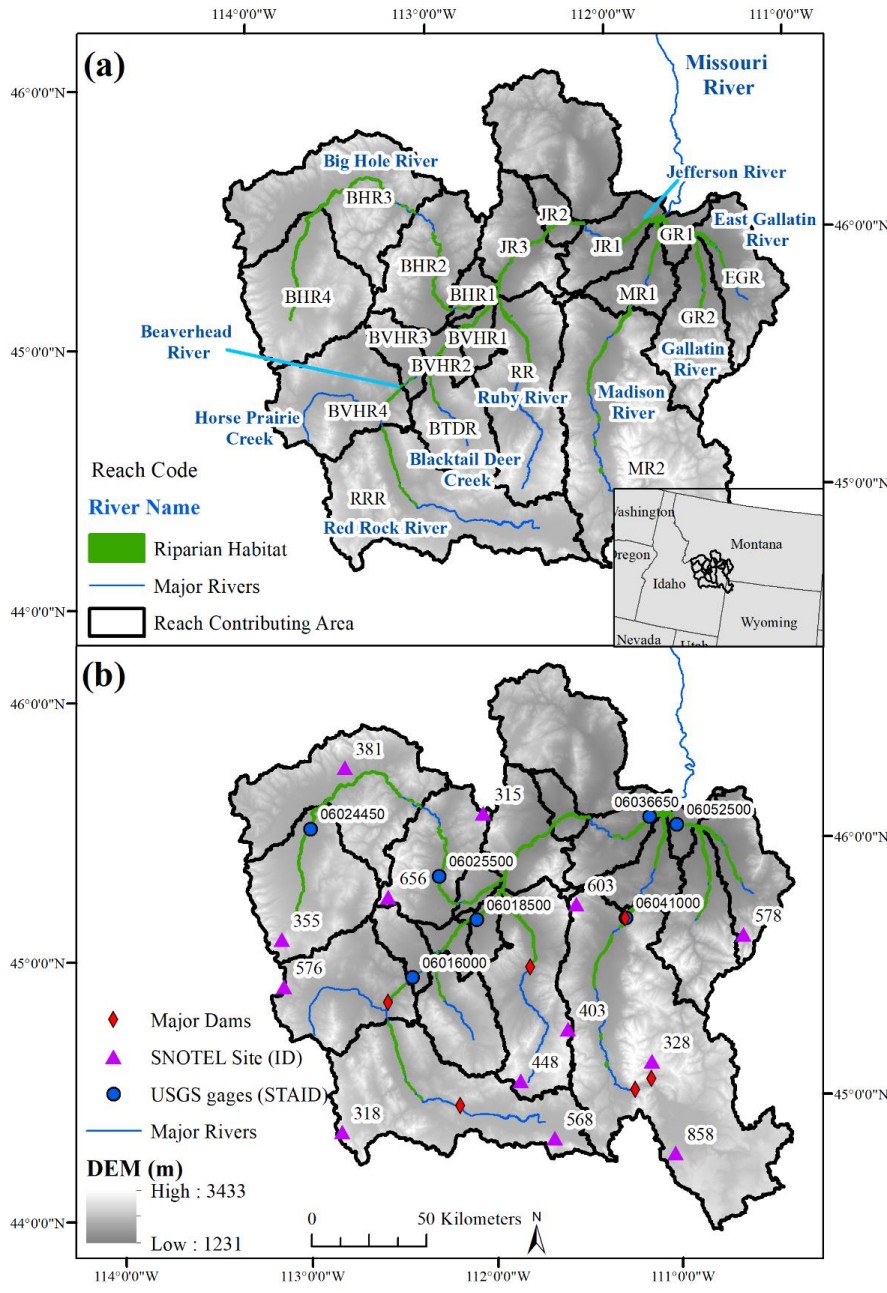

**Figure 1.** (a) The major rivers considered in the analysis, the distribution of the riparian areas evaluated, and the division of the riparian areas into reaches across the Upper Missouri River Headwaters Basin, southwestern Montana, USA. (b) The spatial distribution of the U.S. Geological Survey stream gages and snow telemetry (SNOTEL) sites considered in the analysis. STAID: Station ID, DEM: Digital Elevation Model.



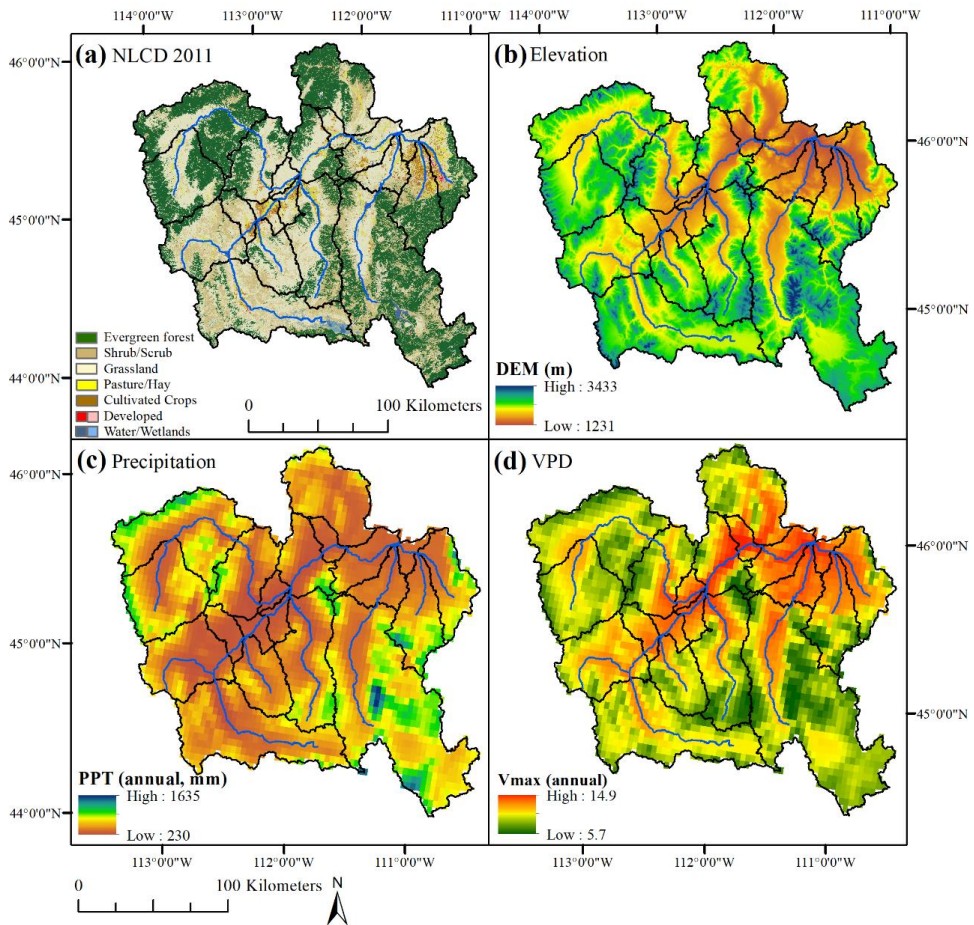

**Figure 2.** Spatial variability in (a) landcover, defined using the 2011 National Land Cover
Database (NLCD), (b) elevation, (c) mean annual precipitation (PPT), and (d) mean annual vapor
pressure deficit (VPD), across the Upper Missouri River Headwaters Basin. DEM: Digital
Elevation Model, Vmax: maximum vapor pressure deficit.

**2.2 Unit of Analysis**

The objective of this study was not to document changes in the total amount of riparian

vegetation, but instead to document temporal variability and trends in the wetness of persistent

riparian vegetation in relation to climate and landscape variables. The extent of persistent

riparian vegetation in major river valleys was delineated manually using Landsat imagery from

1985, 1986, 2016, and 2017 (Table 1) to define our area of analysis. National Agricultural

Imaging Program (NAIP) imagery was also used to improve accuracy in areas where agriculture

was inter-mixed with riparian vegetation. For headwater reaches, riparian areas upstream of all



identifiable irrigated agriculture were excluded from the analysis. For trend analysis, we used

river topology, topography, and clusters of irrigated agriculture to divide the delineated riparian

areas into 19 study reaches (Table 2, Fig. 2). After riparian reach lengths were defined, the per

reach contributing area was calculated using the Spatial Tools for the Analysis of River Systems

(STARS, v 2.0.4) (Peterson, 2017). All pits and flow interruptions in the digital elevation model

(DEM) were filled. The flow direction for the river network was generated and the rivers burned

into the DEM.  The area contributing to the downstream point of each riparian reach (n=19) was

estimated so that each contributing area was non-overlapping with edge-matching inter-basins

(Theobald et al., 2006) (Table 2, Fig. 1).

**Table 1.** Landsat images used to map agricultural extent. The Palmer Hydrological Drought
Index (PHDI) values were provided for the month of July. The percent was calculated based on
the values that occurred between 1984 and 2017. TM: Thematic Mapper, OLI: Operational Land
Imager

| Date | Path/Row | Sensor | PHDI (%) |
|------|----------|--------|----------|
| 6-Aug-85 | p39r28 | TM | -2.85 (12.6) |
| 6-Aug-85 | p39r29 | TM | -2.85 (12.6) |
| 31-Jul-86 | p40r28 | TM | 0.33 (43.0) |
| 31-Jul-86 | p40r29 | TM | 0.33 (43.0) |
| 2-Aug-16 | p40r28 | OLI | -2.22 (19.3) |
| 2-Aug-16 | p40r29 | OLI | -2.22 (19.3) |
| 29-Jul-17 | p39r28 | OLI | -1.03 (35.2) |
| 29-Jul-17 | p39r29 | OLI | -1.03 (35.2) |






**Table 2.** Characteristics of each riparian reach considered including river length, riparian area analyzed, riparian reach contributing area, and average (1984-2016) growing-season (June, July, August, JJA) Normalized Difference Wetness Index (NDWI) and Normalized Difference Vegetation Index (NDVI). Standard error shown in parentheses.

| Reach Code | River | River Length (km) | Riparian Area (ha) | Reach Contributing Area (km²) | Total Upstream Contributing Area (km²) | NDWI (JJA) | NDVI (JJA) |
|---|---|---|---|---|---|---|---|
| JR1 | Jefferson River | 55.4 | 1190 | 1021 | 24711 | 0.17 (0.01) | 0.38 (0.01) |
| JR2 | Jefferson River | 25 | 745 | 395 | 21233 | 0.22 (0.01) | 0.41 (0.01) |
| JR3 | Jefferson River | 48.9 | 1080 | 1348 | 20839 | 0.22 (0.01) | 0.41 (0.01) |
| BVHR1 | Beaverhead River | 47.9 | 805 | 377 | 8867 | 0.20 (0.01) | 0.47 (0.01) |
| BVHR2 | Beaverhead River | 34.3 | 352 | 345 | 8491 | 0.26 (0.01) | 0.51 (0.01) |
| BVHR3 | Beaverhead River | 24 | 218 | 544 | 6774 | 0.21 (0.01) | 0.48 (0.01) |
| BVHR4 | Beaverhead River | 93.8 | 160 | 2236 | 6230 | 0.26 (0.01) | 0.50 (0.01) |
| RRR | Red Rock River | 158 | 410 | 3993 | 3993 | 0.27 (0.01) | 0.50 (0.01) |
| BTDR | Black Tail Deer River | 77 | 26 | 1373 | 1373 | 0.22 (0.01) | 0.45 (0.01) |
| RR | Ruby River | 180.2 | 813 | 2726 | 2726 | 0.27 (0.01) | 0.49 (0.01) |
| BHR1 | Big Hole River | 29.9 | 800 | 317 | 7898 | 0.20 (0.01) | 0.43 (0.01) |
| BHR2 | Big Hole River | 64 | 850 | 1838 | 7581 | 0.23 (0.01) | 0.42 (0.01) |
| BHR3 | Big Hole River | 104.6 | 1623 | 3259 | 5743 | 0.12 (0.01) | 0.37 (0.01) |
| BHR4 | Big Hole River | 75.3 | 1717 | 2484 | 2484 | 0.17 (0.01) | 0.49 (0.01) |
| MR1 | Madison River | 53.7 | 1072 | 886 | 8231 | 0.22 (0.01) | 0.40 (0.01) |
| MR2 | Madison River | 108 | 1771 | 7345 | 7345 | 0.22 (0.01) | 0.38 (0.01) |
| GR1 | Gallatin River | 20.9 | 495 | 310 | 3427 | 0.23 (0.01) | 0.45 (0.01) |
| GR2 | Gallatin River | 54.4 | 1058 | 1660 | 1660 | 0.29 (0.01) | 0.53 (0.01) |
| EGR | East Gallatin River | 73 | 602 | 1457 | 1457 | 0.24 (0.01) | 0.52 (0.01) |





### 2.3 Dependent Variable


The NDWI calculated from Landsat imagery (NIR – SWIR1)/(NIR + SWIR1) (Gao,
1996; McFeeters, 1996) was used to estimate riparian wetness. Relative to other indices such as
the NDVI, NDWI is considered to be less sensitive to atmospheric conditions including solar
elevation angle, sensor angle, and atmospheric condition, making it suitable for time series
analysis (Crétaux et al., 2015), and has been used to monitor patterns in waterlogged areas
(e.g., Chatterjee et al., 2005; Chowdary et al., 2008). NDWI values greater than approximately
0.3 are typically used to distinguish open water (Chatterjee et al., 2005; Chowdary et al., 2008;
McFeeters, 2013). Across the UMH Basin, we determined that riparian NDWI values were more
sensitive to interannual variability in climate (Fig. 3) and river discharge than NDVI, making it a
more appropriate index for this analysis. Per year, average NDWI values (June – August, 1984-
2017, 102 values per riparian reach) were calculated using the Landsat surface reflectance image
collections in Google Earth Engine for all delineated riparian reaches (n=19). June, July and
August were selected to correspond to peak months for irrigation water withdrawals (Bauder,
2018). Potentially erroneous values were defined as values that were greater or less than plus or
minus two standard deviations from the riparian reach-specific mean monthly and were removed.
To normalize the data for seasonal variation values were calculated as the anomaly from the
riparian reach specific, long-term (1984-2017) mean monthly value (NDWI anomaly), then
averaged summer values (June-August) to provide a single NDWI anomaly per summer, per
reach. The multi-month approach compensated for data gaps created when cloud cover masked
Landsat NDWI values.

### 2.4 Independent Variables


Climate variables derived from the Parameter-elevation Regressions on Independent
Slopes Model (PRISM, Daly et al., 2008) included annual precipitation, annual lagged (one year)
precipitation, winter precipitation (January-March), spring precipitation (March-May), summer
precipitation (June-August), spring maximum and minimum temperature (March-May), summer
maximum and minimum temperature (June-August), maximum vapor pressure deficit (VPD;
spring and summer). VPD represents a measure of the drying power of the air and is a function
of air temperature and humidity. Across the contributing area of each riparian reach (n=19), 100
points were randomly selected (total points = 1900). To generate basin-wide values, the climate



values for each year (1984-2016) were extracted for each point, averaged for the reach (Table 3),
then weighted using the relative size (ha) of each reach across the basin. Because upstream
climate, such as snowfall or precipitation, can influence downstream riparian wetness, climate
variables for each riparian reach were similarly calculated using the area-weighted average
values for that reach and all reaches contributing to that reach.
To characterize interannual variability in snowfall, we used a total of 13 Snow Telemetry
(SNOTEL) sites (IDs: 315, 318, 328, 355, 381, 403, 448, 568, 576, 578, 603, 656, 858). Annual
total snowfall (September – August) and total spring snowfall (March-July) were calculated for
each SNOTEL site (Table 3). For each riparian reach we identified the nearest one or two
SNOTEL sites, using the SNOTEL site immediately upstream from the riparian reach as
available. When two SNOTEL sites were used, the snowfall amounts were averaged across the
two sites. Only sites with data available for the entire period of 1984-2017 were used (NSIDC,
2018). To further characterize climate conditions, we included the monthly Palmer Drought
Severity Index (PDSI) and the Palmer Z-Index for NOAA NCDC Division 2 in Montana (Table
3). Both indices are calculated from precipitation and temperature station data and interpolated at
5 km (NOAA NCDC 2014). The PDSI represents the accumulation or deficit of water over the
past approximately 9 months, while the Palmer Z-Index represents the current monthly
conditions with no memory of previous deficits or surpluses (NOAA NCDC 2014). The indices
were averaged to spring (March-May), summer (June-August), and annual, and represent multi-
month averages of the drought indices. Temporal trends (1984-2016) in the climate variables
were tested at the basin scale using the non-parametric Mann-Kendall test for trends (Kendall R
package) (Mann, 1945, Kendall, 1975, Gilbert, 1987). Each SNOTEL site was tested
independently for temporal trends in snowfall.

**2.5 Agricultural Patterns**
We sought to relate patterns in riparian wetness to patterns in total irrigated agricultural
area and the relative abundance of irrigation methods. The USGS Water Use Surveys track
surface and groundwater withdrawals and uses every five years (1950-2015) at a county scale
(USGS 1988; Dieter et al., 2018). In both 1985 and 2015, 99% of water-withdrawals were
surface water, and 99% of the total water withdrawals (surface + groundwater) were for
irrigation across Beaverhead, Gallatin, Jefferson and Madison counties (USGS 1988; Dieter et



al., 2018). Across these counties total water withdrawals were 3% less in 2015 relative to 1985,
although this pattern was variable across the basin with the Gallatin and Madison counties
showing a 27% and 9% increase in water withdrawals, respectively, and the Jefferson and
Beaverhead counties showing a 48% and 15% decrease in water withdrawals, respectively
(USGS, 1988; Dieter et al., 2018). Across the UMH Basin, the Montana Department of
Revenue's Final Land Unit Classification (FLU, 2010 and 2017) provides spatially explicit data
on the irrigation methods used per field, while the U.S. Department of Agriculture's (USDA)
CropScape (2007-2017) provides annual data on the spatial extent and crop type of agriculture.
Between 2010 and 2017, the Montana State's FLU dataset documented a 1.6% increase in total
irrigated agriculture, but a 17% increase in the area irrigated by center pivot irrigation.

These sources of data, however, lacked a spatially explicit dataset of agricultural extent

and irrigation methods for the early part of the Landsat archive (1980s). Therefore, we generated
two agricultural extent datasets representing the two temporal ends of the Landsat archive
(1985/1986 and 2016/2017). The Landsat images used to define the active cropland extent are
shown in Table 1. Cloud cover was only present in the mountainous areas in all images used. We
recognize that by using a single Landsat image (instead of multiple images collected over the
growing-season) and only representing the ends of the study time span, we may be
underestimating agricultural extent and missing year-to-year variability in agricultural activities.
Generating agriculture extent and irrigation types for the beginning and end of our study period,
however, enabled us to identify spatially explicit trends or shifts in agricultural practices that
have been previously shown at a county/state scale (USDA, 2018). Cropland extent was
generated initially using eCognition 9.2 software (Trimble, Westminster, CO). The Landsat
images were segmented into objects using the near infrared (NIR), red, and green bands. The
FLU 2017 data layer was used to mask out non-crop and non-pasture land cover types. The
objects were classified as agriculture or non-agriculture using NDVI thresholds. The draft
agricultural outputs were then manually edited to add and remove agricultural fields as needed.
Fallow fields were not included in the agricultural extent as they were assumed to be non-
irrigated for that year. For overlapping portions between adjacent Landsat images, a field was
included as crop if it was identified as such in either image.





**Figure 3.** A visual comparison of index values in a dry year (2001, 431 mm annual precipitation) and a wet year (1995, 687 mm annual precipitation) at the confluence of Jefferson, Madison and Gallatin Rivers. The Normalized Difference Wetness Index (NDWI) in the riparian vegetation showed more variability in response to precipitation relative to the Normalized Difference Vegetation Index (NDVI). A comparison of (a) NDVI (July 2001), (b) NDWI (July 2001), (c) raw Landsat image (July 1, 2001), (d) NDVI (July 1995), (e) NDWI (July 17, 1995), and (f) raw Landsat image (July 17, 1995). A similar pattern was observed across the basin.






**Table 3.** Climate variables considered in the analysis to represent interannual variability in conditions. The 25th, 50th, and 75th quartile are shown to indicate the variability in the per-riparian reach values included in the random forest (RF) regressions (n=19). The frequency of variable selection for inclusion in the random forest regressions is also shown. When tested at a basin-scale for the time period of 1984-2016, no climate variables showed a significant temporal trend except summer vapor pressure deficit ($* = p < 0.1$). PRISM: Parameter-elevation Regressions on Independent Slopes Model, SNOTEL: snow telemetry, NOAA: National Oceanic and Atmospheric Administration, summer: (June, July, August), spring: (March, April, May)

| Climate Variables | Source | 25th quartile | 50th quartile | 75th quartile | Temporal Trend (tau) | Frequency selected for inclusion in RF regressions |
|---|---|---|---|---|---|---|
| Annual precipitation (mm) | PRISM | 456.1 | 527.1 | 620.4 | -0.03 | 11 |
| 1-year lagged annual precipitation (mm) | PRISM | 458.9 | 532.7 | 625.4 | -0.03 | 2 |
| Precipitation (spring) (mm) | PRISM | 48.1 | 56.2 | 68.0 | -0.004 | 1 |
| Precipitation (summer) (mm) | PRISM | 32.7 | 43.8 | 58.1 | -0.13 | 4 |
| Annual snowfall (snow water equivalent (SWE), mm) | SNOTEL | 938.6 | 1113.4 | 1421.0 | -0.18 - 0.16 | 1 |
| Spring snowfall (March-June) (SWE, mm) | SNOTEL | 169.3 | 264.7 | 402.3 | -0.18 - 0.15 | 7 |
| Maximum temperature (spring) (°C) | PRISM | 9.7 | 11.1 | 12.4 | -0.03 | 3 |
| Maximum temperature (summer) (°C) | PRISM | 23.4 | 24.6 | 25.8 | -0.03 | 1 |
| Minimum temperature (spring) (°C) | PRISM | -4.2 | -3.1 | -2.0 | -0.004 | 0 |
| Minimum temperature (summer) (°C) | PRISM | 5.3 | 6.4 | 7.5 | -0.13 | 0 |
| Vapor Pressure Deficit maximum (spring) | PRISM | 7.1 | 8.1 | 9.0 | 0.07 | 8 |
| Vapor Pressure Deficit maximum (summer) | PRISM | 18.4 | 20.5 | 22.7 | 0.21* | 6 |
| Palmer Z-Index (annual) | NOAA | -0.5 | -0.3 | 0.3 | -0.07 | 9 |
| Palmer Drought Severity Index (annual) | NOAA | -1.6 | -0.2 | 0.8 | -0.11 | 13 |
| Palmer Z-Index (spring) | NOAA | -0.9 | 0.2 | 0.8 | 0.02 | 9 |
| Palmer Drought Severity Index (spring) | NOAA | -1.8 | -0.3 | 1.1 | -0.05 | 8 |
| Palmer Z-Index (summer) | NOAA | -1.5 | -0.4 | 1.0 | -0.15 | 5 |
| Palmer Drought Severity Index (summer) | NOAA | -2.4 | -0.5 | 1.3 | -0.14 | 15 |





Active crop fields were further classified manually as center pivot irrigation or non-center
pivot irrigation (e.g., gravity-fed, non-center pivot sprinkler) based on field shape (i.e., round,
not round). There may be potential confusion between non-center pivot irrigation and non-
irrigated fields, however, 92 and 93% of the 1985/1986 and 2016/2017 agricultural area,
respectively, co-occurred with FLU polygons classified as irrigated, suggesting that non-irrigated
agriculture is a minority cover class across the UMH basin. For reference, the FLU polygons
were classified as center-pivot, sprinkler or gravity-fed using irrigation infrastructure (gates,
ditches, dikes) identifiable from National Agricultural Imaging Program (NAIP) images (1 m
resolution). Sprinkler irrigation was distinguished using parallel wheel lines. Our efforts, in
contrast, did not attempt to distinguish gravity-fed irrigation from non-center pivot sprinkler
irrigation.

**2.6 Analysis**

Temporal trends in riparian wetness (NDWI anomaly ~ Year) were tested for each
riparian reach using the non-parametric Mann-Kendall (MK) test for trends. As the MK test for
trends can be sensitive to temporal autocorrelation (Hamed and Rao, 1998), we used the Durbin-
Watson statistic to test for the presence of temporal autocorrelation in the NDWI anomaly values
of each riparian reach (Table 4). Temporal autocorrelation was found to be significant for the
NDWI anomaly data over time in 3 of the 19 riparian reaches, but in all three cases, the
autoregressive model (AR1) performed worse than the linear model, as evaluated by comparing
Akaike Information Criterion (AICc) values (Hurvich and Tsai, 1989), suggesting that
autoregressive models were not appropriate for this analysis (Table 4). However, because
autocorrelation can inflate trend significance, for these three riparian reaches we calculated a
modified Mann-Kendall test for trends that accounts for the autocorrelation structure of the data
(Hamed and Rao, 1998).
Interannual variability in riparian wetness for a given reach can be expected to be a
function of (1) interannual climate variability and (2) changes in the amount and timing of
anthropogenic water withdrawals or water return flow, while spatial variability in these
relationships can be expected to be a function of landscape characteristics. Temporal variability
in climate and anthropogenic activities could occur both within each reach and upstream of each
reach. Because annual (1984-2016) agricultural and irrigation data were not available for the



entire time series, the influence of water withdrawals was estimated as the residual variance after
modeling the interannual variability in riparian wetness attributable to climate.
The NDWI anomaly values were related to climate variables for each riparian reach using
random forest analysis. The random forest analyses were used to quantify the amount of
variation in the NDWI anomalies explained by climate variables and to identify the frequency
(importance) of particular climate variables in predicting NDWI anomalies. Random forest
techniques use bootstrapping to employ hundreds of regression trees and make no prior
assumptions about cause and effect relationships or correlations among variables (Hastie et al.,
2009). Random forest techniques are generally insensitive to multicollinearity; however, the
inclusion of highly correlated variables can deflate both variable importance and the overall
variation explained by the analysis, while the inclusion of many variables can make
interpretation difficult and introduce noise (Murphy et al., 2010). We therefore implemented
variable selection using the *rfUtilities* package in R (Murphy et al., 2010) before running random
forest regressions for each riparian reach with the selected subset of climate variables. To model
growing-season riparian NDWI anomalies we calculated 500 regression trees for each riparian
reach. Although we did not restrict the number of nodes, model overfit was limited by setting the
minimum sample size per node to 5. Because of the limited data points per riparian reach (n=33)
model fit was assessed using out of bag (OOB) root mean squared error (RMSE, 70% of points
used to train, 30% of points used to validate) using the *randomForest* package in the R statistical
software (Liaw and Wiener, 2015). We found no increase in the OOB error as more trees were
generated (i.e., up to 500 trees). Random forest regression residuals were then extracted and
evaluated for temporal trends not attributable to climate variability (NDWI anomaly random
forest regression residuals ~ Year). Temporal trends in the regression residuals were tested using
the non-parametric MK test for trends. We again used the Durbin-Watson statistic to test for the
presence of temporal autocorrelation in the NDWI anomaly-climate regression residual values of
each riparian reach (Table 4). Temporal autocorrelation was found to be significant for the
residual data over time in 3 of the 19 riparian reaches, so the modified Mann-Kendall test for
trends, that accounts for the autocorrelation structure of the data (Hamed and Rao, 1998), was
used for these three riparian reaches. Differences in agriculture between riparian reaches that
showed a trend in the NDWI anomaly-climate regression residuals and riparian reaches that did
not were compared using the non-parametric Mann-Whitney-Wilcoxon Test.




We note that we tested an alternative method in which data for all riparian reaches and
years were combined in a single linear mixed model. Although this approach increased our
sample size (33 years x 19 riparian reaches), we found that the error in the regression,
specifically the strength of the relationship between the predicted and actual NDWI anomalies,
was uneven between riparian reaches, thereby decreasing our confidence in the analysis of
trends in the residuals. This finding further supported our decision to run a random forest
regression for each riparian reach.

**2.7 Ancillary Spatial Datasets**
Landscape characteristics such as topography, geology, and landcover may influence how
riparian vegetation responds to climate variability over time and were therefore also considered.
Between-group differences in landscape characteristics were calculated for riparian reaches that
showed a temporal trend in riparian wetness relative to riparian reaches that showed no temporal
trend in riparian wetness using the non-parametric Mann-Whitney-Wilcoxon Test (or the
Wilcoxon rank sum test) (Cohen, 1988). Variability in topography was quantified as the (1)
elevation coefficient of variation across each 10-digit hydrologic unit code (HUC-10) (Ascione
et al., 2008), as well as the (2) Melton Ruggedness number, which is calculated as the maximum
elevation minus the minimum elevation divided by the area of the hydrological unit (HUC10)
(Melton, 1965), using the USGS National Elevation Dataset (NED) 10 m resolution (Gesch et
al., 2002). The percent of the riparian reach's within reach contributing area that was (1)
evergreen forest, (2) herbaceous vegetation, (3) pasture, and (4) crop was included, as classified
by the National Land Cover Database (NLCD) 2011 (Homer et al., 2015). Soil and geology
characteristics were considered using the minimum water table depth (April-July), bedrock
depth, and soil drainage characteristics, specifically the percent of each riparian reach's
contributing area that is well drained (excessively drained, somewhat excessively drained, well
drained) and poorly drained (very poorly drained, poorly drained). These variables were derived
from the National Resources Conservation Service's Soil Survey Geographic (SSURGO)
database to characterize infiltration capacity (Soil Survey Staff, 2018). Change in developed
(built-up) land, including urban, residential, and commercial land uses was quantified using the
"Historical built-up intensity layer (1810-2015, 5-year intervals)" (Leyk and Johannes, 2018).
This dataset quantifies the sum of building areas of all structures per pixel, where pixel size is



250 m by 250 m. Change in built-up intensity was quantified as the change in the sum of
building areas between 2015 and 1985 ($m^2$) per river length (m).

**2.8 River Discharge**

Riparian corridors are interconnected with its adjacent rivers via longitudinal, lateral, and

vertical fluxes of water (Fritz et al., 2018). To explore the potential relationship between riparian
water storage and river discharge across the UMH Basin, we identified seven USGS stream
gages within the basin with upstream contributing areas ranging between ~3,400 ha and ~25,000
ha (Table 5). The gages were variable in their position relative to flow regulators such as dams
associated with lakes or reservoirs (Table 5). The amount of flow regulation enforced by these
flow regulators was unknown and therefore a point of uncertainty. The Spearman correlation
coefficient was calculated between the monthly river discharge, averaged to June-August, and
the riparian NDWI for the co-located riparian reach or the riparian reach immediately adjacent to
each gage. We note that a correlation can be indicative of a similar response of both variables to
interannual water availability (e.g., precipitation) as well as potential movement of water across
the river-upland interface. We also evaluated trends in river discharge over time (1984-2016) in
growing-season (June, July, August), as well as autumn (September, October, and November)
and winter (December, January, February) seasons using the MK test for trends. The temporal
trends in river discharge were calculated only to compare with temporal trends in riparian
wetness over the same period. We note that a full trend analysis in river discharge would require
not only utilizing the entire record of river discharge available per gage, but also considering the
potential impact of flow regulation via dams, as well as interannual variability in surface
withdrawals for irrigation, which are closely regulated by Montana State Law (Montana DNRC,

2015).






**Table 4.** Temporal trends in per reach riparian Normalized Difference Wetness Index (NDWI,
June, July, August) anomalies using the Mann-Kendall (MK) test for trends. The Durbin-Watson
(DW) statistic was used to test for the presence of temporal autocorrelation. NDWI anomalies
were modeled against climate variables using random forest regressions. The temporal trends in
the random forest regression residuals were evaluated using MK test for trends. A modification
of the MK (Hamed and Rao, 1998) was used for the reaches where the DW statistic was
significant. RMSE: root mean square error, **: $p<0.05$, *: $p<0.1$

| Reach Code | River | NDWI anomaly DW statistic | NDWI anomaly MK tau | Random forest $R^2$ value | Random Forest RMSE | Residual DW statistics | Residual MK tau |
|---|---|---|---|---|---|---|---|
| JR1 | Jefferson River | 1.56 | -0.22* | 0.65** | 0.02 | 1.74 | -0.28** |
| JR2 | Jefferson River | 2.13 | -0.10 | 0.48** | 0.03 | 2.58 | -0.15 |
| JR3 | Jefferson River | 1.75 | -0.20 | 0.66** | 0.02 | 2.13 | -0.27** |
| BVHR1 | Beaverhead River | 1.51 | -0.35** | 0.53** | 0.03 | 1.36** | -0.27** |
| BVHR2 | Beaverhead River | 1.77 | -0.08 | 0.56** | 0.03 | 1.84 | -0.03 |
| BVHR3 | Beaverhead River | 1.28** | -0.39** | 0.25** | 0.04 | 1.83 | -0.26** |
| BVHR4 | Beaverhead River | 1.40** | -0.36** | 0.47** | 0.04 | 1.51 | -0.36** |
| RRR | Red Rock River | 1.63 | -0.20 | 0.32** | 0.03 | 1.61 | -0.16 |
| BTDR | Black Tail Deer River | 1.57 | -0.35** | 0.48** | 0.04 | 1.87 | -0.30** |
| RR | Ruby River | 1.84 | -0.21* | 0.34** | 0.03 | 2.05 | -0.21* |
| BHR1 | Big Hole River | 1.64 | -0.16 | 0.64** | 0.02 | 1.68 | -0.15 |
| BHR2 | Big Hole River | 2.33 | 0.06 | 0.47** | 0.02 | 2.05 | 0.16 |
| BHR3 | Big Hole River | 2.01 | -0.06 | 0.69** | 0.02 | 2.37 | -0.03 |
| BHR4 | Big Hole River | 2.13 | -0.02 | 0.28** | 0.05 | 2.88** | -0.08 |
| MR1 | Madison River | 2.18 | -0.23* | 0.54** | 0.02 | 2.32 | -0.26** |
| MR2 | Madison River | 2.47 | -0.10 | 0.58** | 0.02 | 2.40 | -0.05 |
| GR1 | Gallatin River | 2.02 | -0.38** | 0.37** | 0.03 | 2.23 | -0.53** |
| GR2 | Gallatin River | 1.97 | -0.16 | 0.23** | 0.02 | 1.68 | -0.09 |
| EGR | East Gallatin River | 2.68* | -0.11 | 0.46** | 0.02 | 2.69* | -0.16 |

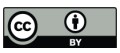



**Table 5.** River discharge characteristics for the U.S. Geological Survey (USGS) gages used in the analysis. Summer (June, July, August) discharge was correlated with the summer Normalized Difference Wetness Index (NDWI) and spring snowfall (March-June) for the riparian reach adjacent to each gage, using the Spearman correlation. Temporal trends were quantified using the Mann-Kendall test for trends. Percent discharge consumed and diverted is from the 2014 Water Plan (MT DNRC, 2014). JJA: June, July, August, SON: September, October, November, DJF: December, January, February, D: dam present at gage, D-US: dam upstream, ND: no dam or minimal flow regulation, na: data not available, SE: standard error, **: $p<0.05$, *: $p<0.1$

| Station ID | USGS Gage Name | Reach Code | Contributing Area (ha) | Consumed (%) / Diverted but not consumed (%) | Flow Regulation | Seasonal mean river discharge (m³ sec⁻¹; ±SE) | | |
| --- | --- | --- | --- | --- | --- | --- | --- | --- |
| | | | | | | Summer (JJA) | Autumn (SON) | Winter (DJF) |
| 6036650 | Jefferson River near Three Forks, MT | JR1 | 24692 | 6% / 20% | D-US | 68.3 (8.3) | 35.0 (2.5) | 33.0 (1.5) |
| 6018500 | Beaverhead River near Twin Bridges, MT | BVHR1 | 8490 | 29% / 69% | D-US | 5.7 (1.7) | 9.0 (1.2) | 8.8 (0.7) |
| 6025500 | Big Hole River near Melrose, MT | BHR2 | 7581 | 13% / 43% | ND | 44.3 (4.5) | 11.4 (0.5) | 10.1 (0.4) |
| 6041000 | Madison River below Ennis Lake near McAllister, MT | MR2 | 7132 | 3% / 11% | D | 56.9 (3.4) | 44.5 (1.5) | 38.5 (0.7) |
| 6016000 | Beaverhead River at Barretts, MT | BVHR3 | 6230 | | D | 20.3 (1.5) | 8.3 (1.2) | na |
| 6052500 | Gallatin River at Logan, MT | GR1 | 3426 | 13% / 37% | ND | 40.7 (3.6) | 18.9 (0.7) | 18.6 (0.4) |
| 6024450 | Big Hole River below Big Lake Creek at Wisdom, MT | BHR4 | 2058 | | ND | 7.9 (1.3) | 1.6 (0.1) | na |

| Station ID | USGS Gage Name | Correlation coefficient (r) | | Seasonal temporal trends (tau) | | |
| --- | --- | --- | --- | --- | --- | --- |
| | | NDWI (JJA) | Snowfall (March-June) | Summer (JJA) | Autumn (SON) | Winter (DJF) |
| 6036650 | Jefferson River near Three Forks, MT | 0.82** | 0.89** | 0.02 | -0.16 | -0.07 |
| 6018500 | Beaverhead River near Twin Bridges, MT | 0.57** | 0.19 | -0.01 | -0.10 | 0.07* |
| 6025500 | Big Hole River near Melrose, MT | 0.60** | 0.84** | 0.12 | 0.07 | 0.16 |
| 6041000 | Madison River below Ennis Lake near McAllister, MT | 0.64** | 0.79** | 0.06 | -0.33** | -0.33** |
| 6016000 | Beaverhead River at Barretts, MT | 0.55** | 0.51** | 0.11 | 0.04 | na |
| 6052500 | Gallatin River at Logan, MT | 0.60** | 0.69** | 0.00 | -0.20* | -0.15 |
| 6024450 | Big Hole River below Big Lake Creek at Wisdom, MT | 0.55** | 0.70** | 0.02 | 0.28** | na |


## 3. Results

### 3.1 Trends in Riparian Wetness

A total of 15,785 ha (157.85 km$^2$) of riparian vegetation was delineated along the major rivers (Fig. 1). River length within each riparian reach ranged from 21 km along the Gallatin River to 180 km along the Ruby River, and averaged 70 km in length (Table 2, Fig. 1). The total riparian area analyzed per reach ranged from 26 ha (289 Landsat pixels) along the Black Tail Deer River to 1771 ha (19,678 Landsat pixels) along the Madison River, and averaged 831 ha (9,233 Landsat pixels, Table 2). The NDVI and NDWI averaged 0.45 and 0.22, respectively, across riparian reaches and years (Table 2). All 19 riparian reaches showed an average NDWI of <0.3 (Table 2), the threshold that is typically used to identify open water (Chatterjee et al., 2005; Chowdary et al., 2008; McFeeters, 2013).

When we tested for MK trends in growing-season (June-August) riparian wetness over time, 8 of the 19 riparian reaches showed a significant decline over time in growing-season NDWI anomalies (5 riparian reaches $p<0.05$, 3 riparian reaches $p<0.1$) (Table 4, Fig. 4). The BVHR3 and BVHR4 riparian reaches that tested positive for autocorrelation still showed a significant drying trend after using the modified MK test. Interannual variability in climate can be expected to explain a portion of the interannual variability in riparian wetness. Across all 19 reaches, climate variables explained 23 to 69% (averaged 47%) of the interannual variability in riparian NDWI anomalies (Table 4). However, basin-wide, the climate variables did not show a temporal trend over same period (1984-2016), apart from the VPD maximum (summer) which showed an increasing trend ($p<0.1$) (Table 3). Drought indices, in particular the PDSI (summer, selected in 15 regressions and annual, selected in 13 regressions), but also the Palmer Z-index (annual and spring both selected in 9 regressions), as well as annual precipitation (selected in 11 regressions) were the variables most frequently selected for inclusion in the random forest analyses (Table 3).

For the eight riparian reaches that showed a temporal trend in NDWI anomalies (Figure 4a) the NDWI anomaly-climate regression residuals also showed a significant negative trend over time, indicating that declines in riparian wetness cannot be attributed solely to climate variability (7 riparian reaches $p<0.05$, 1 riparian reach $p<0.1$, Table 4, Fig. 4b). One additional riparian reach along the Jefferson River (JR3) did not show a significant trend in NDWI anomalies but did show a significant negative trend in the NDWI anomaly-climate regression





residuals ($p<0.05$, Table 4, Fig. 4). The riparian reach BVHR1 also showed a significant negative
trend in the NDWI anomaly-climate regression residuals when tested using the modified MK
test. Data for two of the riparian reaches at the basin outlet (JR1, GR1) are shown in Fig. 5 and
Fig. 6, respectively. Both show a decline in NDWI anomalies over time, with the slope of the
relationship steepening after the removal of the climate component (Fig. 5 and 6).



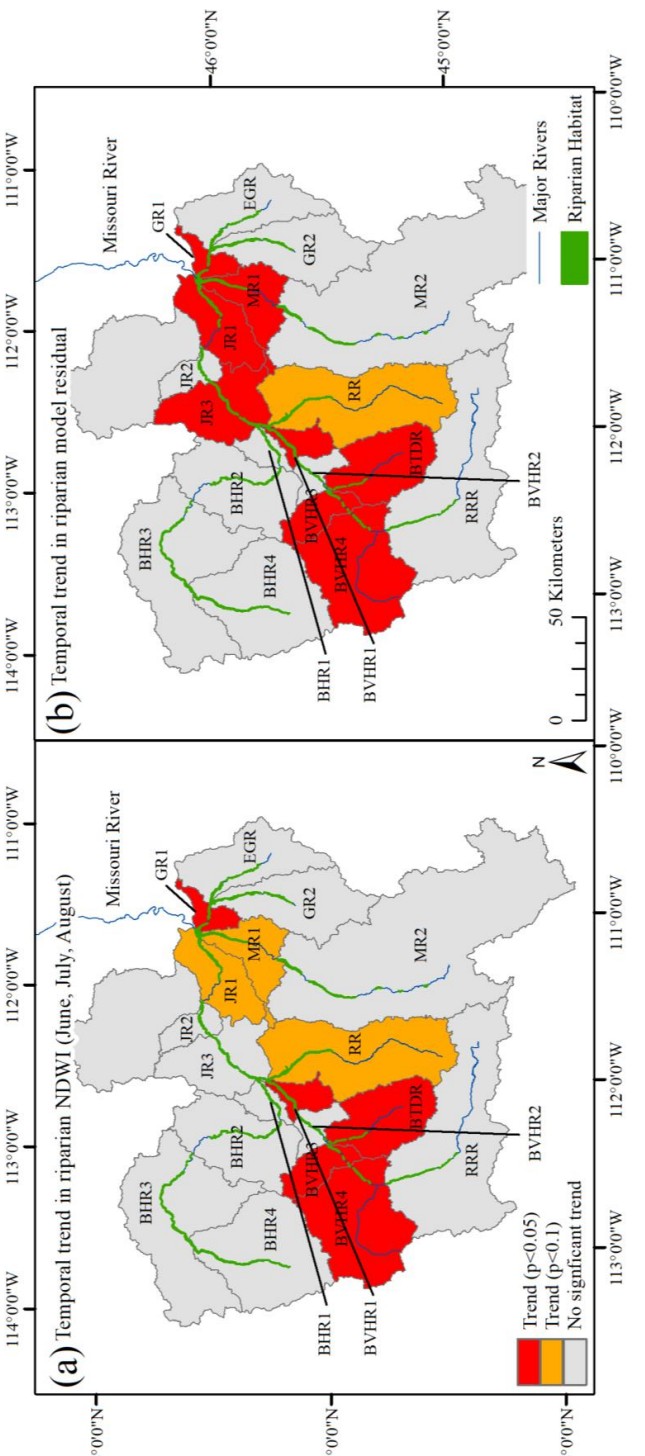

**Figure 4.** (a) The spatial distribution of riparian reaches found to show a significant decreasing trend ($p<0.1$ or $p<0.05$) in riparian wetness using the Normalized Difference Wetness Index (NDWI, June, July, August) anomalies, and (b) the spatial distribution of riparian reaches found to show a significant trend in NDWI anomaly-climate regression residuals, or the variance in NDWI anomalies not explained by climate variables. All trends were negative, indicating a drying over time.




**Figure 5.** Statistics for the Jefferson River riparian reach at the basin outlet (JR1) including, (a)
variability in June, July, August (JJA) river discharge over time (Station ID: 6036650), (b)
relationship between the Normalized Difference Wetness Index (NDWI) and river discharge, (c)
trend in NDWI anomalies over time, (d) correlation between NDWI anomalies and predicted
NDWI anomalies, and (e) trend in NDWI anomalies-climate regression residuals over time.





**Figure 6.** Statistics for the Gallatin River riparian reach downstream of the East Gallatin River
(GR1) including, (a) variability in river discharge over time (Station ID: 6052500), (b)
relationship between the Normalized Difference Wetness Index (NDWI) and river discharge, (c)
trend in NDWI anomalies over time, (d) correlation between NDWI anomalies and predicted
NDWI anomalies, and (e) trend in NDWI anomalies-climate regression residuals over time.



### 3.2 Trends in Agriculture and Water Withdrawals

Agriculture across the UMH Basin is spatially distributed along the major rivers (Fig. 2a). Using the endpoint (1985/86 and 2016/17) agriculture dataset, the largest amounts of agriculture occurred along the Gallatin River, Beaverhead River, Ruby River, and the most upstream reach of the Big Hole River (Fig. 7a). The effect of water withdrawals can be expected to accumulate downstream, therefore the total amount of upstream agriculture was highest along the Beaverhead River, Jefferson River and downstream portion of the Gallatin River (Fig. 7b). Over the study period the total amount of agriculture was relatively stable (4% increase), although we observed a minor decline in total agriculture across the most upstream portion of the basin, and the largest increases in total agriculture along the Gallatin and Jefferson Rivers (Fig. 7 and 8). In contrast, irrigation methods saw much greater changes. We observed a five-fold (506%) increase in the amount of agriculture using center pivot irrigation, and a 39% decrease in the amount of agriculture using non-center pivot irrigation (Table 6). Aerial imagery shows examples of the conversion to center pivot irrigation between 1985 and 2017 (Fig. 8). The percent change in agricultural land area using center pivot irrigation ranged from 0% to 58% across the reaches, with the biggest conversions along the Jefferson, Beaverhead, Madison and Black Tail Deer Rivers (Table 6). After controlling for river length, the Gallatin, Jefferson and Beaverhead saw the largest increases in center pivot agriculture. And the rate of conversion could help explain the drying trends. Riparian reaches that saw a significant decline in riparian wetness, even after accounting for variability explained by climate, showed a 46% average increase in the fraction of reach-scale center pivot irrigation relative to the 32% average increase in riparian reaches where no such temporal trend was observed (Mann-Whitney-Wilcoxon, $p<0.1$). Drying riparian reaches also had more agriculture within and upstream of them than non-drying reaches (269 km$^2$ relative to 171 km$^2$ of active agriculture), but this second difference was not statistically significant (Table 6).

The response of a riparian reach to changes in water withdrawals and irrigation method may also depend on other landscape characteristics such as soil, geology and topography. Riparian reaches that showed a significant non-climate related drying over time showed a higher percent well-drained soils ($p<0.05$) and higher Melton Ruggedness number (greater range in elevation per area, $p<0.05$, Table 7). In addition, although irrigation dominates water consumption across the basin, we note that development has increased around Bozeman, along





the East Gallatin River, over the study period, while minimal increases in development were
found elsewhere (Fig. 7F).

Although the examples in Fig. 5 and Fig. 6 fit the pattern of a shift towards center pivot

irrigation and a corresponding drying trend in riparian wetness, other reaches showed less
intuitive patterns. For instance, all reaches that showed a significant drying trend also showed a
substantial increase in the fraction of center pivot agriculture, ranging from 35% to 64%, except
BVHR4, which showed a significant drying trend without an associated increase in center pivot
agriculture (a 24% increase in center pivot agriculture, but the lowest total ha of center pivot
irrigation in 2016/17 of any riparian reach). The NDWI anomalies and NDWI anomalies-climate
residuals shown in Fig. 9a and 9b indicate that this stretch of the Beaverhead River (BVHR4),
which is immediately downstream from the Clark Canyon Reservoir, experienced a step decrease
in riparian wetness in 2002, with no visible trend before or after 2002. Such a clear step decrease,
however, was not observed in the closest stream gage (Station ID: 06016000) downstream of this
riparian reach. In contrast, one riparian reach on the Beaverhead River further downstream
(BVHR2) showed a 54% increase in the fraction of center pivot agriculture, as well as a decrease
in total agriculture over the study period ($-48.5$ ha km$^{-1}$ river length), with no drying trend (Fig.
9c and 9d), even though reaches upstream and downstream of BVHR2 show significant drying
trends. With the landscape characteristics considered we were again unable to determine why
this riparian reach was more resilient than other riparian reaches of this river.



**Figure 7.** Changes in agricultural and development characteristics across Upper Missouri River Headwaters Basin between 1985/86 and 2016/17 including, (a) total per reach agriculture (2016/17), (b) total agriculture within and upstream of each reach (i.e., accumulated ag) (2016/2017), (c) change to reach-scale abundance of center pivot irrigated agriculture (1985/86 to 2016/17), (d) change to reach-scale abundance of non-pivot irrigated agriculture (1985/86 to 2016/17), (e) change in total per reach agriculture (1985/86 to 2016/17), and (f) change in built-up intensity, defined as the summed building area at 250 m resolution (1985 to 2015).





**Figure 8.** Examples of areas showing a shift in irrigation technique over the past 30 years across
the Upper Missouri River Headwaters Basin including examples at the confluence of the
Beaverhead (center), Big Hole (left), and Ruby River (right), shown in (a) and (c), as well as
examples along Gallatin River shown in (b) and (d).







**Figure 9.** The Beaverhead River (BVHR4) (a) NDWI anomalies over time, (b) NDWI anomalies-climate regression residuals over time, and the Beaverhead River (BVHR2), (c) NDWI anomalies over time, (d) NDWI anomalies-climate regression residuals over time. The MK test for trends was significant ($p<0.05$) for (a) and (b), but not significant for (c) and (d). JJA: June, July, August.




**Table 6.** The per reach abundance of irrigated agriculture (Ag) at the two ends of the time period considered (1985/86 and 2016/17). Irrigation method was identified as center pivot agriculture or non-center pivot agriculture based on field shape. Accumulated ag is defined as the summed area of agriculture across the total contributing area of each reach (e.g., GR1 = agriculture area in GR1, GR2 and EGR). Riparian reaches that showed a significant non-climate related drying over time are shaded gray. Using the Mann-Whitney-Wilcoxon test, the change in fraction center pivot agriculture was significant at the $p<0.1$ level.

| Reach Code | River | Center Pivot Ag (1985/86, ha) | Non-Center Pivot Ag (1985/86, ha) | Center Pivot Ag (2016/17, ha) | Non-Center Pivot Ag (2016/17, ha) | Change in Fraction Center Pivot Ag (%) | Accumulated Ag (1985/86, ha) | Accumulated Ag (2016/17, ha) |
|---|---|---|---|---|---|---|---|---|
| JR1 | Jefferson River | 571 | 2365 | 3444 | 1027 | 58 | 76045 | 78536 |
| JR2 | Jefferson River | 539 | 2544 | 2344 | 1301 | 47 | 71432 | 71661 |
| JR3 | Jefferson River | 601 | 2986 | 3093 | 1998 | 44 | 68349 | 68016 |
| BVHR1 | Beaverhead River | 727 | 9034 | 5631 | 2226 | 64 | 37729 | 34673 |
| BVHR2 | Beaverhead River | 196 | 11794 | 5794 | 4531 | 54 | 27967 | 26815 |
| BVHR3 | Beaverhead River | 810 | 3254 | 3387 | 1772 | 46 | 11774 | 12084 |
| BVHR4 | Beaverhead River | 0 | 1420 | 330 | 1039 | 24 | 1420 | 1368 |
| RRR | Red Rock River | 535 | 5754 | 2368 | 3189 | 34 | 6290 | 5557 |
| BTDR | Black Tail Deer River | 1066 | 3138 | 3351 | 1056 | 51 | 4204 | 4406 |
| RR | Ruby River | 540 | 10414 | 4852 | 5739 | 41 | 10953 | 10591 |
| BHR1 | Big Hole River | 215 | 1780 | 768 | 1029 | 32 | 16080 | 17661 |
| BHR2 | Big Hole River | 0 | 3992 | 1854 | 3789 | 33 | 14085 | 15865 |
| BHR3 | Big Hole River | 52 | 3174 | 83 | 2515 | 2 | 3226 | 2597 |
| BHR4 | Big Hole River | 0 | 6868 | 0 | 7624 | 0 | 6868 | 7624 |
| MR1 | Madison River | 909 | 1445 | 2848 | 1020 | 35 | 9256 | 9451 |
| MR2 | Madison River | 1282 | 5620 | 4128 | 1456 | 55 | 6902 | 5584 |
| GR1 | Gallatin River | 441 | 1957 | 3438 | 1494 | 51 | 14386 | 22717 |
| GR2 | Gallatin River | 221 | 8143 | 4407 | 8133 | 33 | 8364 | 12540 |
| EGR | East Gallatin River | 256 | 3367 | 2175 | 3071 | 34 | 3623 | 5245 |
| **Total** | | **8961** | **89048** | **54294 (506% increase)** | **54006 (39% decrease)** | | **398951** | **412991 (4% increase)** |



**Table 7.** Characteristics of riparian reach contributing areas including median water table depth (m), median bedrock depth (m), percent well-drained (or very well drained) soil, percent poorly (or very poorly) drained soil, elevation coefficient of variation (CV), and Melton Ruggedness number. The Mann-Whitney-Wilcoxon test was used to calculate a measure of the difference (or lack of) between riparian reaches that showed a significant non-climate related drying over time (shaded gray), and riparian reaches that showed no such pattern, with two asterisks indicating a significant difference ($p<0.05$) between the two groups.

| Reach Code | River | Water Table Depth (median) | Bed Rock Depth (median) | Well Drained (%) | Poorly Drained (%) | Elevation CV | Melton Ruggedness Number |
|---|---|---|---|---|---|---|---|
| JR1 | Jefferson River | 84 | 46 | 92 | 3 | 20 | 2.0 |
| JR2 | Jefferson River | 54 | 41 | 87 | 4 | 13 | 3.0 |
| JR3 | Jefferson River | 54 | 36 | 89 | 2 | 22 | 1.4 |
| BVHR1 | Beaverhead River | 54 | 41 | 91 | 3 | 12 | 3.5 |
| BVHR2 | Beaverhead River | 61 | 41 | 81 | 6 | 7 | 2.3 |
| BVHR3 | Beaverhead River | 45 | 46 | 92 | 2 | 15 | 3.0 |
| BVHR4 | Beaverhead River | 80 | 46 | 96 | 2 | 10 | 3.4 |
| RRR | Red Rock River | 15 | 46 | 90 | 4 | 13 | 1.2 |
| BTDR | Black Tail Deer River | 84 | 46 | 91 | 1 | 17 | 3.7 |
| RR | Ruby River | 54 | 48 | 93 | 3 | 20 | 1.9 |
| BHR1 | Big Hole River | 54 | 41 | 99 | 0 | 10 | 3.1 |
| BHR2 | Big Hole River | 31 | 41 | 93 | 2 | 18 | 1.0 |
| BHR3 | Big Hole River | 15 | 38 | 91 | 4 | 13 | 0.8 |
| BHR4 | Big Hole River | 15 | 40 | 86 | 5 | 10 | 1.0 |
| MR1 | Madison River | 46 | 48 | 92 | 4 | 16 | 2.2 |
| MR2 | Madison River | 54 | 64 | 60 | 2 | 15 | 0.3 |
| GR1 | Gallatin River | 46 | 41 | 92 | 3 | 11 | 3.0 |
| GR2 | Gallatin River | 84 | 48 | 84 | 3 | 24 | 1.3 |
| EGR | East Gallatin River | 84 | 41 | 83 | 3 | 21 | 1.3 |
| Mann-Whitney-Wilcoxon p-value | | 0.45 | 0.37 | 0.04** | 0.21 | 0.51 | 0.02** |




**3.3 Trends in River Discharge**

Growing-season riparian NDWI was significantly correlated ($p<0.05$) with growing-season river discharge at all seven USGS stream gages analyzed (Spearman correlation coefficient ranged between 0.55 along Beaverhead River and Big Hole River and 0.82 along the Jefferson River) (Table 5). In addition, all gages, except the Beaverhead River at Twin Bridges gage, were significantly correlated with spring snowfall (Spearman p-value <0.05), the climate variable that showed the highest correlation on average between summer discharge and the climate variables considered in the analysis. Unlike the riparian reaches, we saw no temporal trend (1984-2016) in the growing-season river discharge for any of the seven gages evaluated. However, because the watershed is a snowmelt-driven system, we also tested if trends were restricted to the low-flow seasons (autumn and winter). During the autumn months (September, October, November) we observed a decline in river discharge at the Madison River ($p<0.05$) and Gallatin River ($p<0.1$) gages and an increase at the Big Hole River gage near Wisdom ($p<0.05$), which is near the upstream end of the Big Hole River (Table 5). During the winter months (December, January, February) we observed a decline in river discharge at the Madison river gage ($p<0.05$) and an increase in river discharge at the Beaverhead River near the Twin Bridges gage ($p<0.1$) (Table 5).

**4. Discussion**

Across the western U.S., water withdrawals, diversions and impoundments associated with agriculture have contributed to riparian degradation (Goodwin et al., 1997; Klemas, 2014). In examining the multi-decadal trends in riparian wetness for a total of 158 km$^2$ of riparian ecosystem across the UMH Basin, we found long-term, significant drying along 8 of the 19 riparian reaches in this basin, including all three of the riparian reaches (the Jefferson, Madison and Gallatin Rivers) at the confluence forming the Missouri River. In contrast, we did not observe trends in growing-season river discharge or climate variables over the same period. The persistence of drying trends in riparian vegetation after accounting for the influence of climate variability, and the correlation of riparian drying with a basin-wide shift in agricultural irrigation practices, suggest that the complexities of agricultural water use and crop management are likely to be contributing factors to the drying of riparian areas in this basin. Water withdrawals across the basin are almost entirely surface-water (99%) and for irrigation (99%) (USGS 1988; Dieter et



al., 2018). The agricultural data generated in this study indicate that the basin has experienced a
substantial shift from non-center pivot irrigation (e.g., gravity-fed or sprinkler) to center pivot
irrigation (from 9% to 50% center pivot irrigation, basin wide). This shift in irrigation practices
is concentrated along the Beaverhead, Jefferson and Gallatin Rivers, all of which showed
statistically significant drying in at least portions of their riparian reaches. Correspondingly, the
Big Hole River sub-watershed, which is dominated by gravity-fed irrigated hay and pasture
(Montana DNRC, 2014), showed the least amount of conversion to center pivot irrigation
relative to other sub-watersheds over the study period, with no temporal trends in riparian
wetness.
Similar shifts in irrigation methods have occurred across the western U.S., where the
percentage of agricultural land irrigated by sprinkler systems, including center pivot irrigation,
increased from 28% to 59% between 1984 and 2013 (Schaible, 2017). Advances in irrigation
technology allow for water to be applied at the most appropriate timing in plant root zones to
increase crop consumptive use of water and therefore, crop yields (Falkenmark and Lannerstad,
2005; Ward and Pulido-Velazquez, 2008). However, despite the shift to more efficient irrigation
methods, the total water-use for irrigation across the U.S. remained largely stable over the same
period (Schaible, 2017). This patterns may indicate that local water savings do not necessarily
translate to the watershed scale. Increases in crop yields are linearly correlated with increases in
evapotranspiration (Steduto et al., 2012), so that the reduction in water application is often off-
set by increases in evapotranspiration, specifically crop transpiration (Ward and Pulido-
Velazquez, 2008; Grafton et al., 2018). A schematic of the potential impact of irrigation method
on water cycling is shown in Fig. 10. Further, proposed water savings in per field water
applications often fail to account for farm-level decisions and incentives (Ward and Pulido-
Velazquez, 2008). Within the current water rights framework, more efficient water use can
incentivize farmers to make changes to crop choices and crop rotation patterns, or to increase the
total area irrigated or the frequency of irrigation so that their water rights and usage are
maintained and maximized (Pfeiffer and Lin, 2014; Grafton et al., 2018). If there is a local
reduction in water usage downstream water users can more fully exercise their water rights so
that there is no net reduction in water usage at the watershed scale (Ward and Pulido-Velazquez,

2008).





Riparian and river condition for a given reach can be expected to be a function of its
upstream river network, including water added and removed from upstream reaches, as well as
upstream land uses (Ver Hoef and Peterson, 2012; Fritz et al., 2018). Biotic integrity, for
example, has been shown to depend on upstream conditions (Schofield et al., 2018), which can
extend tens of kilometers up the channel network (Van Sickle and Johnson, 2008). In
consideration of this, the climate variables used to model temporal variability in riparian wetness
were calculated as a function of each reach's total upstream contributing area. Additionally, we
considered upstream accumulated changes, such as the upstream accumulated agriculture, to help
interpret trends in the NDWI anomaly-climate regression residuals. Cumulative effects of both
climate and land use may explain why the basin's three most downstream riparian reaches (on
the Gallatin River, Madison River, and Jefferson River) all saw significant drying trends in the
NDWI anomaly-climate residuals, or the NDWI anomalies after accounting for climate
variability. The incremental drying effect might also help explain why we did not observe
temporal trends in riparian wetness in some headwater riparian reaches. For instance, along the
headwater riparian reaches of the Madison River (MR2), the Gallatin River (GR2), as well as the
East Gallatin River (EGR), the analyzed riparian vegetation extended to the upstream end of
irrigated agriculture. Although the total amount of agriculture varies among these riparian
reaches, potentially the incremental drying effects of irrigation on groundwater storage and
return flow  do not become evident (spectrally or hydrologically) until accumulated lower in the
watershed. In addition to water use, landscape characteristics can inform how a riparian
ecosystem responds to changes in reach- or basin-scale hydrology. Well-drained soils and a
higher Melton Ruggedness number, characteristics significantly associated with the reach-scale
riparian drying trends, can be expected to facilitate the return flow of excess irrigation water to
the riparian corridor. Implying that a shift towards more "water efficient" irrigation might have a
greater drying effect on nearby riparian vegetation.
While the presence of riparian drying trends in the NDWI anomaly-climate residuals
indicated that the observed drying trends were not solely attributable to climate, climate
variability was a significant predictor of the interannual variability in riparian wetness (e.g., Fig.
5 and Fig. 6), a finding documented in other geographic regions as well (e.g., Fu and Burgher,
2015; Nguyen et al., 2015; Huntington et al., 2016). Drought events, and the resilience of river
and riparian ecosystems to these events, are a significant concern for  stakeholders in the Upper



Missouri Headwaters Basin (Montana DNRC, 2015; McEvoy et al., 2018).  Although evaluation
of water rights and corresponding water withdrawals under drought conditions was beyond the
scope of this study, our findings suggest that the conversion to center pivot irrigation could
amplify the impacts of reduced precipitation on riparian areas. Additionally, an increasing
summer VPD could further increase crop water losses to evapotranspiration (Massmann et al.,
2018), potentially exacerbating both the hydrological effect and salinization effect of irrigation
conversion (Singh, 2015). We note, however, that climate and river discharge trends were
quantified only to be compared with trends observed in riparian wetness over the same period
(1984-2016). Because only partial climate and river discharge records were used, our findings
regarding the presence or absence of trends in the climate and river discharge data should be
interpreted with caution.

Despite only partial discharge records being utilized, one interesting finding was that

over the same period a drying trend in riparian areas did not necessarily translate into a trend in
river discharge. We can speculate that because the rivers are snow-melt dominated (Markstrom
et al., 2016; Cross et al., 2017), during the summer months irrigation return flow may have an
impact on riparian areas but could represent a relatively small percent of summer flows. A
comprehensive water budget or hydrological modeling approach, however, would be needed to
quantify this, and specifically to determine how anthropogenic activities may have a differential
impact on riparian wetness relative to river discharge. Additionally, rivers across the basin vary
in the amount of flow regulation from dams. For example, the Big Hole River and Gallatin
Rivers are relatively unregulated while the Madison River, Beaverhead River, Ruby River and
Red Rock River are all regulated by dams. The reservoirs above dams retain water during the
spring runoff, reducing peak flows, and release more water in the autumn, changing a river's
natural flow regime (Montana DNRC, 2014). It is possible that shifts in dam management and
corresponding changes in flow regulation could contribute to trends in riparian wetness.
However, river discharge (JJA) was significantly correlated with spring snowfall at eight of nine
gages, suggesting that even with seasonal flow regulation, discharge along dammed rivers still
typically represents interannual variability in climate.

Efforts to characterize the factors influencing variability and trends in riparian wetness

are critical to maintain and restore riparian functionality. Healthy floodplains and riparian areas
serve a number of functions including slowing runoff, promoting local groundwater recharge,



and quickening the recovery of local groundwater storage post-drought (Montana DNRC, 2014).
Spectral indices calculated from satellite imagery have been successfully used to monitor the
response of riparian vegetation to variability in channel morphology (Henshaw et al., 2013;
Hamdan and Myint, 2015), as well as changes induced by the installation of in-stream restoration
structures (Hausner et al. 2018; Vanderhoof and Burt, 2018). While Landsat has been commonly
used to examine multi-decadal trends in vegetation condition (Goetz et al., 2005; McManus et
al., 2012; White et al., 2017), because of the narrow, linear footprint of riparian ecosystems
within human-influenced landscapes, efforts to apply Landsat time-series analysis to riparian
systems have been limited (e.g., Henshaw et al., 2013; Hamden and Myint, 2015; Nguyen et al.,
2015). Regional-scale Landsat efforts have tended to focus on changes to riparian extent rather
than riparian trends in greenness or wetness (e.g., Jones et al., 2010; Macfarlane et al., 2017).
Along river systems, however, the moderate resolution of Landsat can misrepresent riparian
edges or fail to detect portions of the riparian corridor that are narrower than Landsat's minimum
mapping unit, potentially influencing the calculated spectral patterns. In our analysis we
minimized such errors by (1) restricting the analysis to rivers with riparian corridors large
enough to be measured using Landsat, and (2) using a consistent riparian area extent across the
time series. It is clear, however, that finer spatial resolution sources of imagery will be critical
for riparian corridors too narrow to be monitored with Landsat imagery. To this end, data sources
with increased spatial resolution are rapidly becoming more available and useful for monitoring
water resources (e.g., Sentinel-2, CubeSats) (e.g., Vande Kamp et al., 2013; Gärtner et al., 2016;
Cooley et al., 2017; Yang et al., 2017), but lack the multi-decadal data records provided by
Landsat. This means that for larger riparian corridors, Landsat spectral indices remain a critical
data source that can be used to characterize trends in riparian wetness as well as potentially
quantify the impact of land use changes, including long-term shifts in irrigation methods, on
riparian vegetation.





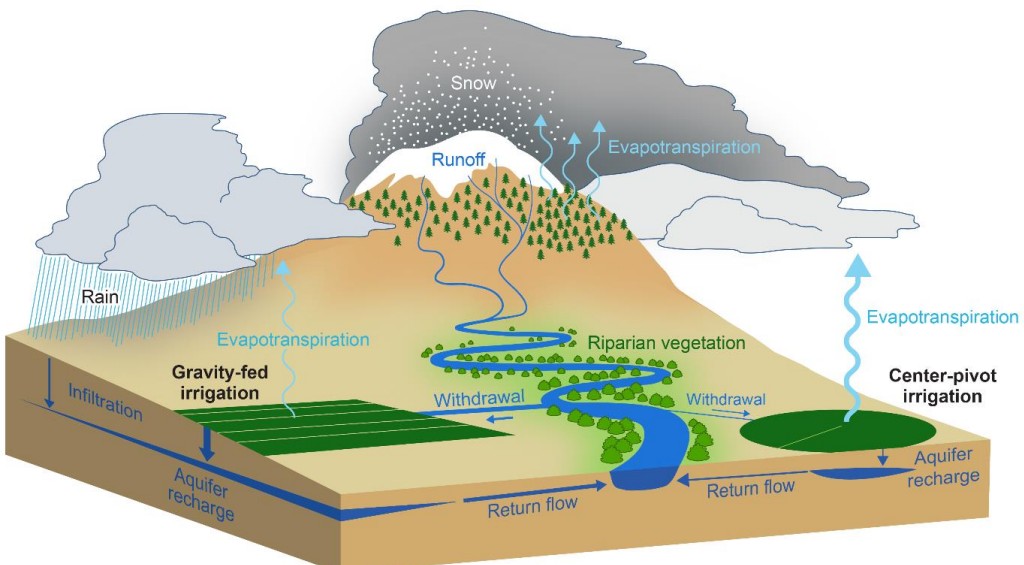

**Figure 10.** A schematic showing the potential impacts of changing irrigation types. While
shifting to center-pivot irrigation can be expected to reduce per-field water applications, it can
also be expected to increase evapotranspiration as well as decrease sub-surface return-flow and
aquifer recharge. Reduced withdrawal may not persist downstream but instead be used by the
same farmer or a downstream user. Thicker and thinner lines are used to indicate more or less
water, respectively.

**5. Conclusion**
Riparian corridors provide valuable ecosystem functions including storing water,
nutrients, pollutants, and sediments, providing wildlife corridors, and influencing water
temperature (Vivoni et al., 2006; Lees and Peres, 2008; Isaak et al., 2012). A drying trend in
riparian areas across the Upper Missouri Headwaters Basin could lessen the effectiveness of
these functions and shift the systems towards more drought-tolerant plant species that are less
adapted to highly variable flow regimes (Capon, 2013; Catford et al., 2014). Although promoted
as a more water-efficient approach, several recent studies have demonstrated a lack of
catchment-scale water savings after farmers shift towards center pivot irrigation (Perry, 2017;
Grafton et al., 2018). We were able to pair a Landsat time series analysis with climate and
agricultural data to document a statistically significant drying trend, not explained by climate
variability, along nearly half (42%) of riparian reaches in the Upper Missouri Headwaters Basin.
Although the riparian reaches experiencing drying trends tended to have more upstream
agriculture and greater shifts toward center pivot irrigation, the correlations between agricultural





activities and riparian wetness were imperfect, suggesting that the upstream river network, as
well as other reach-scale characteristics such as the riparian species or the geology/soil
characteristics, also influence the response of a riparian reach to changes in water withdrawal. In
addition, the drying trends in riparian ecosystems were not observed in the snow-melt driven
river discharge (JJA), a finding that should be explored further using hydrological models.
Maintaining and improving riparian functionality across watersheds dominated by agricultural
activity will require not only more efforts to track temporal trends in riparian vegetation, but also
more efforts to separate out the relative influence of climate and anthropogenic activities.

**6. Acknowledgements**
This project was funded by the U.S. Geological Survey, Land Resources, Land Change Science
Program as well as by a U.S. EPA Region 8 grant, entitled "Building drought resiliency and
watershed prioritization using natural water storage techniques" and through the associated
interagency agreement (DW-014-92475401-0). We thank Haley Distler for her assistance in
delineating the riparian corridor and Jeremey Havens for his assistance in generating the
hydrology schematic. We also thank Ken Fritz and Robert Payn for their insightful comments on
earlier versions of this manuscript. Following publication, the data related to this publication will
be published in the U.S. Geological Survey's ScienceBase catalog
(https://doi.org/10.5066/P976LZ2G). Any use of trade, firm, or product names is for descriptive
purposes only and does not imply endorsement by the U.S. Government. This publication
represents the views of the authors and does not necessarily reflect the views or policies of the
U.S. EPA.

**7. Author Contributions:** MV, JC, and LA designed the study, MV and JC derived the input
datasets, MV performed the analysis, and MV, JC, and LA wrote the manuscript.

**8. Competing interests:** The authors declare that they have no conflict of interest.

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
