# Peer review of "Influence of multi-decadal land use, irrigation practices and climate on riparian corridors"

_Hydrology and Earth System Sciences, 2019_

## Referee Comment (RC1) · Richard marinos (Referee) · 16 May 2019

Review submitted as PDF for nicer formatting.

Note to editors: It would be nice if this web form could have text formatting, other than LaTeX!

I think this is a nice study. The authors used some clever methods to infer how changes in irrigation practices might be altering riparian zone wetness in semi-arid regions of the Missouri basin. They do a great job of synthesizing a large number

of disparate datasets. The analyses are thoughtful, the results are interesting, and the discussion is comprehensive. The authors are careful to note caveats and do not make statements that outstrip the evidence. The manuscript would have been much stronger if the authors had shown how center-pivot irrigation trends changed over time, rather than just using the two endpoints in the analysis. Then the authors could have used a joint model that included climate and land use, rather than this two step, regression-on-residuals approach. I have some philosophical issues with doing regressions on residuals, especially when the explained variation from the climate model varies widely between basins. Doing this would require rewriting the whole paper, though, and I don't think this is a fatal flaw by any means. I have some questions and minor quibbles that I hope the authors can address in revisions. I recommend minor revisions and look forward to seeing the responses of the authors. -Richard Marinos Line Comments: Lines 81, 92, 111: Minor stylistic point; you lead each paragraph with qualifiers (e.g. "Although...") which can obscure the main thrust of the paragraph. Line 135: "Our research questions included"... could you list all the research questions that this paper includes? Else, just say that these were your two questions. Figure 2: Did you derive these P and VPD data yourself using the PRISM model, or are these available data products that you used? If the former, please include this in the results of your paper, not the methods. Line 183: It seems to me that this approach, only looking at the riparian vegetation that persisted during the study period, introduces an issue of survivorship bias. Can you justify this choice further in light of this critique? Line 185: Did you use the DEM to inform identification of riparian vs. upland vegetation? Did you exclude the active channel from your analyses? Line 190: Could you briefly expand on how you arrived at these specific reaches, either in comments or in the manuscript itself? It seems from the map that contiguous riparian areas cross the boundaries of your reaches. What distinguishes them as units of analysis? Line 228: I wonder how correlated cloud cover and higher NDWI values are, and if this would skew the analysis toward lower NDWI values. Though you did say that most P is as snowpack. Not really much to be done about this anyway, just musing.

Lines 281-299: How well does this imagery analysis mesh with the cropland extent in the NLCD? Figure 3: This was very helpful in understanding your data resolution with respect to riparian zone size. Line 357: I am trying to work through the statistical implications of letting the input climatic variables for the random forests vary by reach. I would feel more confident if you could explain more why you took this approach, rather than using the same variables across reaches. Line 391: This CV approach seems strange to me, unless your datum was the lowest point in the HUC unit. Is this what you did? Otherwise a HUC unit at a mean elevation of 100 feet would have 10x the CVof the exact same HUC unit if it was transported to a mean elevation of 1000 feet. Line 417: Saying it's an uncertainty is an understatement! Ok but I see you've qualified your uses of this more in the following lines. Table 5: Why is only March-June snowfall considered? Did I miss something? Methods general comment: You present a LOT of results in your Methods section. I'd prefer to see these moved to the Results section. Figures 5 and 6: These are good figures that answered a lot of questions for me. Could you include as a supplement these plots for all reaches? I'd be interested to know what the "messier" reaches look like. Line 518: I know you give this in Table 6, but could you provide absolute areal changes here too? It's hard to interpret these percentages without knowing absolute area as well. Figure 8: Nice, love these pics. Line 651: I am having a hard time understanding this point about cumulative effects... unless your ratio of recharge areas (e.g. mountains with snowpack) to withdrawal areas becomes smaller with basin size, in which case I could see how this could be the case. Line 688: Appreciate this strong caveat.

Please also note the supplement to this comment:
https://www.hydrol-earth-syst-sci-discuss.net/hess-2019-137/hess-2019-137-RC1-supplement.pdf

---

## Referee Comment (RC2) · Anonymous Referee #2 · 18 Jul 2019

HESS Reviewer Comments

Line 28: would be helpful to specify what "non-center pivot irrigation" includes earlier in the paper (perhaps including in the abstract). There is some discussion of this on lines 321-325. The lack of distinction between gravity fed irrigation and non-center pivot sprinkler irrigation seems significant. Authors should indicate what is known about the efficiency/consumptive water use rates of non-center pivot sprinkler vs. center-pivot vs. flood. It is my understanding that non-center pivot sprinkler would be much more similar to center-pivot (than to flood) in terms of efficiency/consumptive water use. If

non-center pivot sprinkler is not separated out from flood irrigation, authors need to be very clear and specific about what this study tells us about flood/gravity fed irrigation.

Line 50: what is "ditching"? Please re-phrase or clarify

Line 129-131: These citations might be as good or better to make the point that there is increased interested in river resiliency:

Montana Drought Demonstration Partners, 2015: A Workplan for Drought Resilience in the Missouri Headwaters Basin: A National Demonstration Project. http://dnrc.mt.gov/divisions/water/management/docs/surface-water-studies/workplan_drought_resilience_missouri_headwaters.pdf (Accessed May 20, 2019).

Montana DNRC, 2014: Upper Missouri Basin: Water Plan 2014. http://dnrc.mt.gov/divisions/water/management/docs/state-water-plan/upper-missouri/river-basin-plan/upper_missouri_basin_report_final.pdf (Accessed May 29, 2019). Montana DNRC, 2015: Montana State Water Plan: A Watershed Approach to the 2015 Montana State Water Plan. 80. The citation for McEvoy et al 2018 which is used later in the paper also supports this point – specifically for UMH - and summarizes the goals of the MT Drought Demonstration Project Table 2 & Lines 225-228. As a social scientist familiar with the issue and study region, my strength is not in the technical aspects of remote sensing or hydrology, so please take this comment/question with a grain of salt. I am a bit confused as to why authors report the "average NDWI" and "average NDVI" in table 2 given that they are more interested in trend over time (not average). The text on lines 225-228 perhaps explains this – but the paragraph focuses on the per summer "anomaly" rather "average". Also this text does not refer back to figure 2. Greater explanation of why authors report the average in Table 2 would be helpful. In general, the description of the use the anomaly seems more complicated than it needs to be (?). Lines 321-325: please see my earlier comment re: lack of distinction between non-center pivot sprinkler and flood

irrigation. Authors should include a comment on line 325 about whether/how this lack of distinction effects the results – and more importantly what it allows the authors to conclude about flood/gravity fed irrigation practices. Line 328: the use of the "$\sim$" symbol in "NDWI $\sim$ Year" is not clear to me. If the use of "$\sim$" is standard in the field, then ignore my comment, otherwise please specify what that means. This comment might be related to my previous comment about use of "average NDWI" and "average NDVI" in Table 2 and the explanatory text re: use of "anomaly" on Lines 225-228. Line 374: the phrase "differences in agriculture" seems to be missing a modifier or unit. Is it difference in "agricultural area" or in "agricultural practices"? Please specify what this difference is within agriculture that is referred to. Line 515: the phrase "total amount of agriculture was relatively stable" – should specify the ag unit authors are referring to (I assume this is acres of land in agricultural production? But could be ag output/yield, which could mean an increase in ag productivity on same amount of land or stable output, but on fewer acres). Line 554: same comment as above for phrase "decrease in total agriculture over they study period" – specify unit of ag (acres? Or production/output/yield?) Line 667 – same comment "..total amount of agriculture [add units]" Line 519-520. Would be helpful if authors can explain how center-pivots get implemented on the ground. If center pivots increase by 506%, but non-center pivots only decrease by 39% where are these newly added center pivots going? Are they not replacing non-center pivot? Are they replacing flood irrigation at a rate of greater than 1:1? Are they being added to newly expanded agricultural fields (this is not allowed under MT DNRC's water rights laws, which require irrigators to specify place of withdrawal – and specifies that there should not be an expansion of irrigated acreage when irrigators switch to new irrigation system – though this most certainly happens.) Figure 7: I believe the headings in c&d should read "Change to reach-scale pivot irrigation" (not "agriculture"). Figure 7: use of term "built-up" and "building area" in both figure and the associated text is confusing. I assume authors are referring to urbanization, but that is not clear. Line 618: why use the word "crop management"? I expected authors to state: "complexities of ag water use and irrigation practices (or

methods)". In my mind, "crop management" refers to things like change which type of crop is grown, fallowing, use of cover crops, timing of planting and harvesting, etc. Line 636: phrase "total water-use for irrigation across the US" should be more specific. Following Perry et al's 2017 recommendation, authors should specify whether they are referring to water withdraws or water consumption (the following discussion illustrates this point using ET, but it seems like the authors could be more careful/specific with their use of the word "water-use" in line 636. Line 670 "water use" – again, authors should be more specific. Is this "water withdraws"? or irrigation methods? Or general water use – if so, specify some examples of what this includes Line 636-650 Perry et al 2017 make this same point at the global scale. Seems like their paper should be cited in this part of the discussion.

---

## Author Comment (AC1) · 13 Aug 2019

Response to Comments

Manuscript: Influence of multi-decadal land use, irrigation practices and climate on riparian corridors across the Upper Missouri River Headwaters Basin, Montana

Authors: Melanie K. Vanderhoof, Jay R. Christensen, Laurie C. Alexander

Reviewer #1:

[Figure]

Summary Comment: I think this is a nice study. The authors used some clever methods to infer how changes in irrigation practices might be altering riparian zone wetness in semi-arid regions of the Missouri basin. They do a great job of synthesizing a large number of disparate datasets. The analyses are thoughtful, the results are interesting, and the discussion is comprehensive. The authors are careful to note caveats and do not make statements that outstrip the evidence. The manuscript would have been much stronger if the authors had shown how center-pivot irrigation trends changed over time, rather than just using the two endpoints in the analysis. Then the authors could have used a joint model that included climate and land use, rather than this two step, regression-on-residuals approach. I have some philosophical issues with doing regressions on residuals, especially when the explained variation from the climate model varies widely between basins. Doing this would require rewriting the whole paper, though, and I don't think this is a fatal flaw by any means. I have some questions and minor quibbles that I hope the authors can address in revisions. I recommend minor revisions and look forward to seeing the responses of the authors. -Richard Marinos Response: We appreciate the supportive comments provided by Richard Marinos. We agree that the analysis would be stronger if we had spatially explicit, annual data on irrigation methods and abundance. Because the analysis involved a large number of datasets, generating an additional 30 years of agriculture data was beyond scope. However, we hope that the findings presented in the analysis provide motivation either for our research group or for others to generate more agricultural datasets that include data on irrigation type. We have addressed all questions and quibbles below.

Line Comments: Lines 81, 92, 111: Minor stylistic point; you lead each paragraph with qualifiers (e.g. "Although. . .") which can obscure the main thrust of the paragraph. Response: We have removed the term "although" from the start of paragraphs as recommended.

Line 135: "Our research questions included". . . could you list all the research questions that this paper includes? Else, just say that these were your two questions. Response: We have revised this phrase to clarify that those were our 2 research questions.

Figure 2: Did you derive these P and VPD data yourself using the PRISM model, or are these available data products that you used? If the former, please include this in the results of your paper, not the methods. Response: We did not derive these variables ourselves. The P and VPD data were from the PRISM model dataset as specified at the start of section 2.4.

Line 183: It seems to me that this approach, only looking at the riparian vegetation that persisted during the study period, introduces an issue of survivorship bias. Can you justify this choice further in light of this critique? Response: I think what this comment is getting at, is that if a reach had experienced a severe drying trend then riparian vegetation may have transitioned to non-riparian vegetation (e.g., grassland) which would then be missed by the analysis. We focused on persistent riparian vegetation for two reasons. First, evaluating temporal trends while changing the riparian extent from year to year introduces the possibility of conflating temporal change with spatial change. Second, agriculture tends to be immediately adjacent to, and particularly further from the outlet, is often in the riparian area. Focusing on persistent wetland vegetation allowed us to avoid areas within riparian areas that went in and out of active agricultural activity. To address this comment we added the following sentence to section 2.2. "This approach enabled us to reduce uncertainty in the temporal analysis and increase our confidence in the vegetation type but limited our ability to detect changes in riparian extent induced by climate or changes in human land use."

Line 185: Did you use the DEM to inform identification of riparian vs. upland vegetation? Did you exclude the active channel from your analyses? Response: A 30 m DEM was found to be inadequate to separate riparian from agricultural and upland vegetation, therefore we did not use it in the delineation. Yes, the active channel was excluded from the area of analyses. We have added a comment to that effect.

Line 190: Could you briefly expand on how you arrived at these specific reaches, either in comments or in the manuscript itself? It seems from the map that contiguous riparian areas cross the boundaries of your reaches. What distinguishes them as units of analysis? Response: We first used the confluences of rivers or the entrance of major tributaries to divide rivers into reaches. As the reaches were still quite long at this point, we then used the distribution of agriculture, which tended to occur in clusters along the major rivers, so that breaks between clusters of agriculture were used as further dividing points. Future work should focus on moving the analysis to a pixel-scale analysis, eliminating the need for deriving distinguishable reaches.

Line 228: I wonder how correlated cloud cover and higher NDWI values are, and if this would skew the analysis toward lower NDWI values. Though you did say that most P is as snowpack. Not really much to be done about this anyway, just musing. Response: It is an interesting thought! Yes, in this watershed the snowpack is the major driver of river discharge, therefore I suspect the influence of cloud cover would play a relatively minor role.

Lines 281-299: How well does this imagery analysis mesh with the cropland extent in the NLCD? Response: We did not compare the multiple sources of crop data with the NLCD. The NLCD provides land cover data only every 5 years and provides no specific data on crop type or irrigation method.

Figure 3: This was very helpful in understanding your data resolution with respect to riparian zone size. Response: Thank you.

Line 357: I am trying to work through the statistical implications of letting the input climatic variables for the random forests vary by reach. I would feel more confident if you could explain more why you took this approach, rather than using the same variables across reaches. Response: All reaches considered the same set of climate variables. Our goal with this decision was to find the "best fit" between the independent climate variables considered, and the dependent variable. Past efforts (e.g., Murphy et

al., 2010) have found variable selection to improve random forest models. Ecologically, it makes sense that the best fit climate variables may change slightly as we move from snow pack mountains down to the Basin outlet. We also note that many of the climate variables were highly correlated with each other, so a statistical selection of one variable over another, may have modified the model very little.

Line 391: This CV approach seems strange to me, unless your datum was the lowest point in the HUC unit. Is this what you did? Otherwise a HUC unit at a mean elevation of 100 feet would have 10x the CV of the exact same HUC unit if it was transported to a mean elevation of 1000 feet. Response: The elevation coefficient of variation was calculated as the elevation standard deviation divided by the mean elevation, not as the mean elevation. As you can see in Table 7, we do not see a directional trend in the elevation coefficient of variation as we move up the watershed.

Line 417: Saying it's an uncertainty is an understatement! Ok but I see you've qualified your uses of this more in the following lines. Response: In addition to the qualifications, we added the word "major" to the phrase "point of uncertainty."

Table 5: Why is only March-June snowfall considered? Did I miss something? Methods general comment: Response: We considered both annual and spring snowfall. Both are listed in Table 3. In our analysis, spring snowfall consistently out-performed annual snowfall and was one of the best single predictors to represent annual climate and water availability for this Basin.

Comment: You present a LOT of results in your Methods section. I'd prefer to see these moved to the Results section. Response: We moved the supplementary agriculture statistics to the Discussion section and moved the 3 tables that contained results data to the Results section.

Figures 5 and 6: These are good figures that answered a lot of questions for me. Could you include as a supplement these plots for all reaches? I'd be interested to know what the "messier" reaches look like. Response: Providing all of the graphs for all plots

would add a lot of extra pages! The key statistics for each reach are currently provided in Table 3. We have provided (attached) the graphs for our "messiest" reach (defined as the lowest random forest R2 (GR2) Gallatin River below. We hope that this adequate.

Line 518: I know you give this in Table 6, but could you provide absolute areal changes here too? It's hard to interpret these percentages without knowing absolute area as well. Response: We added the absolute areal change values.

Figure 8: Nice, love these pics. Response: Thanks!

Line 651: I am having a hard time understanding this point about cumulative effects. . . unless your ratio of recharge areas (e.g. mountains with snowpack) to withdrawal areas becomes smaller with basin size, in which case I could see how this could be the case. Response: We substantially shortened this paragraph to limit the discussion of cumulative effects. We did retain the sentences explaining the need to look at impact of upstream changes and conditions on the downstream reach of interest.

Line 688: Appreciate this strong caveat. Response: Thank you.

 

Graphs for our "messiest" reach (defined as the lowest random forest R$^2$ (GR2) Gallatin River.

[Figure]

**Fig. 1.**

---

## Author Comment (AC2) · 13 Aug 2019

Response to Comments

Manuscript: Influence of multi-decadal land use, irrigation practices and climate on riparian corridors across the Upper Missouri River Headwaters Basin, Montana

Authors: Melanie K. Vanderhoof, Jay R. Christensen, Laurie C. Alexander

Reviewer #2:

Line 28: would be helpful to specify what "non-center pivot irrigation" includes earlier in the paper (perhaps including in the abstract). There is some discussion of this on lines 321-325. Response: We revised the abstract and no longer use the term non-center pivot irrigation until the Methods section. In the methods section we now expand our description to specify "(e.g., gravity-fed, non-center pivot sprinklers such as tower sprinklers, solid set and permanent sprinklers, side roll, big gun or traveler, or hand move sprinklers)"...

Comment: The lack of distinction between gravity fed irrigation and non-center pivot sprinkler irrigation seems significant. Authors should indicate what is known about the efficiency/consumptive water use rates of non-center pivot sprinkler vs. center-pivot vs. flood. It is my understanding that non-center pivot sprinkler would be much more similar to center-pivot (than to flood) in terms of efficiency/consumptive water use. If non-center pivot sprinkler is not separated out from flood irrigation, authors need to be very clear and specific about what this study tells us about flood/gravity fed irrigation. Response: In response to this comment we added to the Methods that, "Because this irrigation infrastructure was not visible in the Landsat imagery, we did not attempt to distinguish gravity-fed irrigation from non-center pivot sprinkler irrigation. Consequently the datasets as created enabled us to quantify changes in irrigation extent and shifts toward center-pivot irrigation. It did not allow us to make estimates of water consumption or quantify shifts from gravity-fed irrigation to non-center pivot sprinkler irrigation." We also added a paragraph to the Discussion to directly respond to this comment: "One source of uncertainty in our analysis is that at the Landsat scale we were unable to confidently distinguish gravity-fed irrigation from non-center pivot sprinkler irrigation, methods of irrigation that can be expected to show different rates of water efficiency. This source of uncertainty made it difficult to reach definitive conclusions about reach-scale changes in the consumptive water use using our data alone. However, our assumption of a transition away from gravity-fed irrigation and towards center-pivot irrigation is consistent with other comparable sources of data. Across Montana the FRIS surveys (1984 and 2013) documented an increase in the fraction irrigated with center

pivot from 9% to 30%, a decrease in the fraction irrigated with gravity-fed irrigation from 77% to 57%, and a minimal change (<3%) in the fraction of agriculture irrigated with non-center pivot sprinklers (USDA, 1985, 2014). Across the UMH Basin, the Montana Department of Revenue's Final Land Unit Classification (FLU) surveys documented a 17% increase in center-pivot irrigation and a corresponding decrease in both sprinkler and gravity-fed irrigation between 2010 and 2017. Despite these ancillary datasets, however, it is possible that shifts from gravity-fed irrigation to non-center pivot sprinkler irrigation, have also contributed to changes in return flow and riparian condition."

Line 50: what is "ditching"? Please re-phrase or clarify Response: We revised this to "drainage and water diversion ditches".

Line 129-131: These citations might be as good or better to make the point that there is increased interested in river resiliency: Montana Drought Demonstration Partners, 2015: A Workplan for Drought Resilience in the Missouri Headwaters Basin: A National Demonstration Project. http://dnrc.mt.gov/divisions/water/management/docs/surface-waterstudies/workplan_drought_resilience_missouri_headwaters.pdf (Accessed May 20, 2019).

Montana DNRC, 2014: Upper Missouri Basin: Water Plan 2014. http://dnrc.mt.gov/divisions/water/management/docs/state-water-plan/uppermissouri/river-basin-plan/upper_missouri_basin_report_final.pdf (Accessed May 29, 2019). Montana DNRC, 2015: Montana

State Water Plan: A Watershed Approach to the 2015 Montana State Water Plan. 80. The citation for McEvoy et al 2018 which is used later in the paper also supports this point – specifically for UMH - and summarizes the goals of the MT Drought Demonstration Project Table 2 & Lines 225-228. Response: We agree, the citations suggested are a better fit to justify this sentence then the original citations. We have replaced the citations as recommended.

Comment: As a social scientist familiar with the issue and study region, my strength

is not in the technical aspects of remote sensing or hydrology, so please take this comment/question with a grain of salt. I am a bit confused as to why authors report the "average NDWI" and "average NDVI" in table 2 given that they are more interested in trend over time (not average). The text on lines 225-228 perhaps explains this – but the paragraph focuses on the per summer "anomaly" rather "average". Also this text does not refer back to figure 2. Greater explanation of why authors report the average in Table 2 would be helpful. In general, the description of the use the anomaly seems more complicated than it needs to be (?). Response: Table 2 was meant to provide an overview of reach-specific characteristics. Inherent spectral differences between reaches could contribute to our understanding of why we might see variability in the trends between reaches. We have added this sentence in response to this comment. "Reach-scale average NDVI and NDWI values were provided to give a sense of the reach-scale variability in spectral characteristics (Table 2)." In response to the second part of the comment, NDVI has been much more widely used relative to NDWI for the analysis of riparian areas. For this reason we felt it was important to justify our decision.

Lines 321-325: please see my earlier comment re: lack of distinction between non-center pivot sprinkler and flood irrigation. Authors should include a comment on line 325 about whether/how this lack of distinction effects the results – and more importantly what it allows the authors to conclude about flood/gravity fed irrigation practices. Response: Please see the responses above and the text added to the Methods and Discussion sections. We also note that we substantially revised how the ancillary agriculture datasets are presented so that the statistics can act in direct complement to the data generated within this study.

Line 328: the use of the "âĹij" symbol in "NDWI âĹij Year" is not clear to me. If the use of "âĹij" is standard in the field, then ignore my comment, otherwise please specify what that means. This comment might be related to my previous comment about use of "average NDWI" and "average NDVI" in Table 2 and the explanatory text re: use of "anomaly" on Lines 225-228. Response: We have removed the symbol "∼" for

increased clarity.

Line 374: the phrase "differences in agriculture" seems to be missing a modifier or unit. Is it difference in "agricultural area" or in "agricultural practices"? Please specify what this difference is within agriculture that is referred to. Response: We deleted this sentence as we found it a bit out of place here.

Line 515: the phrase "total amount of agriculture was relatively stable" – should specify the ag unit authors are referring to (I assume this is acres of land in agricultural production? But could be ag output/yield, which could mean an increase in ag productivity on same amount of land or stable output, but on fewer acres). Response: We revised this to specify hectares of land in agricultural production. We also want to note that we caught an error in that the percent change in irrigated area had been mistakenly calculated from the accumulated irrigated area, not the per-reach irrigated area. When we calculated the change correctly we found a 10.5% increase in irrigated area. We added a secondary source to the Discussion that found at the state level an increase of 19% in total hectares of irrigated area over a similar period.

Line 554: same comment as above for phrase "decrease in total agriculture over they study period" – specify unit of ag (acres? Or production/output/yield?) Response: We revised this to "total hectares of irrigated agriculture". We did not attempt to calculate product, output or yield, just total area growing crops.

Line 667 – same comment "..total amount of agriculture [add units]" Response: We revised to avoid the term "total amount" throughout and instead specified "hectares".

Line 519-520: Would be helpful if authors can explain how center-pivots get implemented on the ground. If center pivots increase by 506%, but non-center pivots only decrease by 39% where are these newly added center pivots going? Are they not replacing non-center pivot? Are they replacing flood irrigation at a rate of greater than 1:1? Are they being added to newly expanded agricultural fields (this is not allowed under MT DNRC's water rights laws, which require irrigators to specify place of withdrawal

– and specifies that there should not be an expansion of irrigated acreage when irrigators switch to new irrigation system – though this most certainly happens.) Response: In response, we changed the way the irrigation statistics were presented to improve clarity. So percent change, of course, depends on the value you started with (percent change = (post – pre) / pre *100 and because there was very little center pivot irrigation in the mid-1980s our percent change values were large. We now specify the total number of ha and present the relative percent of center pivot and non-center pivot. So center-pivot irrigation went from 9% of irrigated area (8961 ha) to 50% of irrigated area (54,295 ha). We saw primarily conversion from non-center pivot to pivot irrigation, but we also observed land changing from not actively cultivated to center-pivot irrigation. Particularly along the Gallatin River.

Figure 7: I believe the headings in c&d should read "Change to reach-scale pivot irrigation" (not "agriculture"). Response: Caption changed as recommended.

Figure 7: use of term "built-up" and "building area" in both figure and the associated text is confusing. I assume authors are referring to urbanization, but that is not clear. Response: The dataset is called "built-up intensity" which is defined as the summed building area at 250 m resolution. We modified the caption to best match the language used in the figure.

Line 618: why use the word "crop management"? I expected authors to state: "complexities of ag water use and irrigation practices (or C3 methods)". In my mind, "crop management" refers to things like change which type of crop is grown, fallowing, use of cover crops, timing of planting and harvesting, etc. Response: Wording was changed as recommended.

Line 636: phrase "total water-use for irrigation across the US" should be more specific. Following Perry et al's 2017 recommendation, authors should specify whether they are referring to water withdraws or water consumption (the following discussion illustrates this point using ET, but it seems like the authors could be more careful/specific with

their use of the word "water-use" in line 636. Response: This is a good point. We used "total water use" because this was the term used to label the data in the graph in Schaible (2017). To clarify we used the figure caption which specified "total water applied for irrigation"

Line 670: "water use" – again, authors should be more specific. Is this "water withdraws"? or irrigation methods? Or general water use – if so, specify some examples of what this includes Response: We removed the term "water use" here.

Line 636-650: Perry et al 2017 make this same point at the global scale. Seems like their paper should be cited in this part of the discussion. Response: We added references to the Perry et al. (2017) paper to this paragraph.

———————————————————